# Exploring the role of bedrock representation on plant transpiration response during dry periods at four forested sites in Europe

César Dionisio Jiménez-Rodríguez[1], Mauro Sulis[1], and Stanislaus Schymanski[1]

[1]Environmental Research and Innovation (ERIN) Department, Luxembourg Institute of Science and Technology (LIST), Belvaux, L-4422, Luxembourg.

**Correspondence:** César Dionisio Jiménez-Rodríguez (cesar.jimenez@list.lu)

**Abstract.** Forest transpiration is controlled by the atmospheric water demand, potentially constrained by soil moisture availability, and regulated by plant physiological properties. During summer periods, soil moisture availability at sites with thin soils can be limited, forcing the plants to access moisture stored in the weathered bedrock. Land surface models (LSMs) have considerably evolved in the description of the physical processes related to vegetation water use but the effects of bedrock position and water uptake from fractured bedrock has not received much attention. In this study, the Community Land Model version 5.0 (CLM 5) is implemented at four forested sites with relatively shallow bedrock and located across an environmental gradient in Europe. Three different bedrock configurations (i.e., default, deeper, and fractured) are applied to evaluate if the omission of water uptake from weathered bedrock could explain some model deficiencies with respect to the simulation of seasonal transpiration patterns. Sap flow measurements are used to benchmark the response of these three bedrock configurations. It was found that the simulated transpiration response of the default model configuration is strongly limited by soil moisture availability at sites with extended dry seasons. Under these climate conditions, the implementation of an alternative (i.e., deeper and fractured) bedrock configuration resulted in a better agreement between modeled and measured transpiration. At the site with a continental climate, the default model configuration accurately reproduced the magnitude and temporal patterns of the measured transpiration. The implementation of the alternative bedrock configurations at this site provided more realistic water potentials in plant tissues but negatively affects the modeled transpiration during the summer period. Finally, all three bedrock configurations did not show differences in terms of water potentials, fluxes, and performances on the more northern and colder site exhibiting a transition between oceanic and continental climate. Model performances at this site are low, with a clear overestimation of transpiration compared to sap flow data. The results of this study call for increased efforts into better representing lithological controls on plant water uptake in LSMs.

**Keywords:** sap flow, transpiration, weathered bedrock, soil texture, plant hydraulic traits.

# 1 Introduction

Bedrock structure and composition influence the ecosystem productivity (Hahm et al., 2014; Jiang et al., 2020) through its effect on plant nutrient and water uptake (Ding et al., 2021). Physical and chemical weathering processes allow the formation of cracks in the bedrock (Pope, 2015), increasing the presence of water reservoirs that are commonly neglected in ecological analyses at sites characterized by thin soils and seasonal droughts (Sternberg et al., 1996). These reservoirs are highly dynamic, storing water from winter precipitation (Vrettas and Fung, 2017) and sometimes holding even more moisture than the overlying soil (Jones and Graham, 1993; Rempe and Dietrich, 2018). The thickness of the weathered bedrock varies according to rock type and climatic conditions; some European sites with gneiss, granite, and flysch rock types have mean thicknesses of 7.0 m, 4.6 m, and 3.4 m, respectively (Šamonil et al., 2020). Plants growing in seasonally dry environments rely on deep water reservoirs often available within the weathered bedrock (Barbeta et al., 2015; Ding et al., 2021; Sternberg et al., 1996). As an example, vegetation growing in karst systems are able to access water stored in the fractured bedrock thanks to the presence of sediments and organic matter within the cracks of the first 2 m of soil that increase the water holding capacity of the karst formation (Phillips et al., 2019; Querejeta et al., 2007; Swaffer et al., 2014). This suggests that rock moisture is likely an important water reservoir supporting dry season water use at sites with shallow soils. Therefore, inclusion of this additional water reservoir in models may be important for correctly simulating vegetation water use, especially under climate change, where seasonal changes in intensity and frequency of precipitation and drought (Hosseinzadehtalaei et al., 2020; Grillakis, 2019) are expected to result in extended drought severity and duration. However, due to the limited field data, detailed knowledge about the structure of weathered bedrock and its effect on the overlying vegetation remains poorly understood.

Plant transpiration is driven by atmospheric water demand and controlled by stomata, where water diffuses away from leaves along the same pathway as $CO_2$ enters the leaves before it is fixed through the photosynthesis. Water transpired by leaves needs to be replenished by root water uptake and suction-driven transport from roots to leaves. The plant hydraulic system enabling this transport is vulnerable to embolism under strong suction (e.g. low soil water potential), therefore plants reduce stomatal conductance and transpiration under such conditions (Kirkham, 2014; Sperry et al., 1998, 2002). Trees developed a number of strategies that enhance and guarantee the access to different water sources depending on the environmental conditions. Their root system can extend horizontally over large areas (e.g, *Abies* sp., *Meterosideros* sp.) or vertically into the ground (e.g, *Quercus* sp., *Pinus* sp.) reaching depths of more than 20 m (Pallardy, 2008). During prolonged dry periods, some tree species use these extended root systems to access deep groundwater reservoirs (Barbeta et al., 2015; Zhang et al., 2018), redistribute water in superficial soil layers through hydraulic lift (Alagele et al., 2021; Bayala and Prieto, 2020), and/or source water from the weathered and fractured bedrock (Ding et al., 2021; Querejeta et al., 2007; Sternberg et al., 1996; Swaffer et al., 2014). This latter water pool (i.e., moisture extracted by roots from the weathered bedrock) is poorly quantified at the global scale and especially for forested ecosystems that are widespread on hillslopes with thin soils (Jiang et al., 2020; Schwinning, 2010).

Land surface models (LSMs) have evolved considerably over the last decades and have been established as useful tools to understand ecosystem responses to water stress conditions and heat waves (Fisher and Koven, 2020). However, these developments have increased the number of parameters exerting control over multiple fluxes, highlighting the need of providing a better way to constrain LSMs simulations (Mu et al., 2021). These constraints are of high importance because they allow assessing the effect of the parameter selection on the model sensitivity and uncertainty (Lawrence et al., 2020). The representation of subsurface processes plays a critical role in the chemical, thermal, and hydrological fluxes simulated by LSMs (Choi et al., 2013; Davison et al., 2015). This representation, which varies quite strongly between LSMs, has been advanced over the recent years (e.g., Felfelani et al. (2021)) requiring additional constraints to accurately estimate water and energy fluxes. For instance, the Joint UK Land Environment Simulator (JULES) LSM has a 10 m soil column with 28 soil layers (with 14 layers distributed over the first 3 m of soil). This soil column can be shallower depending on the site conditions, where the bedrock is considered only for temperature exchange and not for groundwater storage (Chadburn et al., 2015). NOAH LSM includes an unconfined aquifer beneath a soil column of 2 m depth allowing the soil column to drain freely (Niu et al., 2011). The Organising Carbon and Hydrology In Dynamic Ecosystems (ORCHIDEE) LSM has a total soil depth of 2 m, allowing multiple soil layer discretizations (Campoy et al., 2013), with the bedrock defining a discontinuity in the soil zone processes (Sun et al., 2021). Finally, the Community Land Model version 5.0 (CLM 5) implements a spatially distributed soil thickness (Swenson and Lawrence, 2015) within a range of 0.4 m to 8.5 m depth, derived from a spatially explicit soil thickness data product (Pelletier et al., 2016). Water stored in the unsaturated zone is under negative pressure and considered as the only source of plant water by the model. Depending on the soil water balance, a saturated zone builds up on top of the bedrock but this is a temporal water storage from where any water excess moves out as drainage. Overall, LSMs are based on the common assumption that soil water is the main water source for the vegetation, and the deep drainage is used only for river and/or wetland flow routing and hence neglected as a potential source of water. This conceptualization results in a critically low storage capacity for plant available water at sites characterized by thin soil and prolonged dry periods. Furthermore, due to the lack of detailed field observations for accurately determining the depth-to-bedrock (DTB), this parameterization leads to a large uncertainty in applying LSMs at such sites (Shangguan et al., 2017).

Recent field evidence documents vegetation dependency on rock moisture during dry periods or extended droughts (Carrière et al., 2020; Klos et al., 2018; Hahm et al., 2020; Jiang et al., 2020; Nardini et al., 2021; Qi et al., 2018; Rempe and Dietrich, 2018), and the dependence of maximum rooting depths on the in-situ hydrological regulation (Fan et al., 2017). However, consideration of rock moisture in current models is scarce and the few examples are linked to model rock weathering processes (Cipolla et al., 2021) leaving aside its role for root water uptake and plant transpiration (Fan et al., 2019). From a pragmatic modelling perspective, the inclusion of this additional source of water could be achieved either by implementing a bottomless soil column allowing the exponential root profile to access deep soil water (de Rosnay and Polcher, 1998) or deepening the DTB and/or altering the soil composition for the bedrock layers. This latter approach emphasizes the key role played by soil texture and rock fragments in regulating the response of root growth (Hu et al., 2021; Li et al., 2019) and plant hydraulics to soil drying (Cai et al., 2021; Carminati and Javaux, 2020) as well as the effect of uncertainty in determining the DTB (Brunke

et al., 2016). Recent studies stressed the importance of the accumulated sediments in weathered bedrock fractures that enhances the water holding capacity beneath the soil (Jiang et al., 2020). These sediments allow the woody ecosystems to use additional moisture across different rock types (McCormick et al., 2021). The widespread use of rock moisture by vegetation is supported by the vast number of studies carried out during the last two decades stressing the importance of adding the weathered bedrock as an additional water reservoir that serves as a buffer water supply during dry conditions (Aranda et al., 2007; Baldocchi et al., 2010; David et al., 2013; Forner et al., 2018; Gil-Pelegrín et al., 2017; Hubbert et al., 2001; Penuelas and Filella, 2003; Nadezhdina et al., 2008; Zapater et al., 2012). This reservoir could be critical for properly representing transpiration fluxes during dry periods (e.g., summer, drought, heat waves) of vegetation growing in water-limited environments. Consequently, it is necessary to provide a preliminary evaluation of the impact that additional water storage may have on the simulated transpiration fluxes.

The objective of this work is to evaluate the impact of different bedrock conceptualizations on the simulation of plant transpiration response using CLM 5. The underlying hypothesis of the work is that the omission of plant available water stored in the weathered bedrock could explain some of the model deficiencies in reproducing seasonal transpiration patterns. It is expected that during summer, the vegetation has a fully developed canopy exhibiting a peak in leaf area index, extracting soil water at maximum rate and relying on deeper soil water pools soil water pools during extended dry periods. To test this hypothesis, CLM 5 is implemented at four forested sites located across an environmental gradient in Europe using three different bedrock configurations, (i) the model default, (ii) a deeper location of the impermeable layer, and (iii) a weathered bedrock overlying the impermeable bedrock. The simulated plant transpiration of the three model configurations is compared to the up-scaled transpiration signal measured by sap flow sensors at the selected study sites. Finally, the plant physiological implications of using different bedrock configurations are assessed by examining the plant vulnerability curves of the xylem and leaf segments of the simulated plants.

## 2 Study Sites

Figure 1a illustrates the spatial distribution of DTB across a domain covering a large part of the European continent; 43 % of the area with DTB between 0 m and 2 m, 40 % between 2 m and 40 m, and 17 % with depths larger than 40 m (Pelletier et al., 2016). This geological pattern is overlaid by three main climatic zones according to the Köppen-Geiger climate classification: (i) warm temperate climate with dry summers (Cs) surrounding the Mediterranean sea, (ii) warm temperate fully humid climate (Cf) located in the central part of Europe, and (iii) a snow fully humid climate (Df) covering Eastern Europe and the Scandinavian Peninsula (Beck et al., 2018). The four study sites were selected from the SAPFLUXNET data set (Poyatos et al., 2021) in order to sample the dominant geological and climate settings of the European continent (Table 1). These sites have DTB no larger than 1.5 m covering the shallow soils in mountain ranges (ES-Alt), foothills (FR-Pue, FR-Hes), and lowlands (RU-Fyo). All the sites lack detailed information concerning the physical properties of the weathered bedrock, such as hydraulic conductivity,

water holding capacity, or percentage of fractures. The sites are distributed across an environmental gradient (of mean annual precipitation: 400-900 mm yr$^{-1}$, and air temperature: 4.7-13.8 °C) where the FR-Pue and ES-Alt sites exemplify the Cs climate class of the Mediterranean basin, FR-Hes the Cf class of Central Europe, and RU-Fyo the Df class of Eastern Europe.

## 2.1 Spain, Alto Tajo [ES-Alt]

This research site is located in the Alto Tajo Natural Park (40.8044° N-2.2328° W). The main soil types are classified as calcaric cambisols, mol!cleptosols, and rendzic leptosols (Zapico et al., 2017). These soils are formed from Cretaceous carbonate rocks settled on top of sandy sediments (Carcavilla et al., 2008), with a poor soil development (Granda et al., 2012), and a thickness ranging from 25 cm up to 1.0 m depth (Martín-Moreno et al., 2014). The soils are formed from Cretaceous carbonate rocks settled on top of sandy sediments (Carcavilla et al., 2008), showing a poor development (Granda et al., 2012) with maximum soil thickness varying between 25 cm at the top of the slopes, and more than 1.0 m at the bottom (Martín-Moreno et al., 2014). The climate is classified as continental Mediterranean, with a summer precipitation characterized by high-intensity rainstorms (Martín-Moreno et al., 2014), and snow fall during winter (Acuña Míguez et al., 2020). The site registered a mean annual temperature and precipitation of 11.7 °C and 567 mm yr$^{-1}$, respectively (Poyatos et al., 2016). The vegetation is characterized by four types of forest communities dominated by *Juniperus thurifera* L., *Pinus nigra* J.F. Arnold ssp. *salzmannii* (Dunal) Franco, *Quercus faginea* Lam., and *Quercus ilex* ssp. *ballota* (Desf) Samp. (Forner et al., 2014). *Quercus ilex* trees at this site have the capacity to allocate structural and fine roots down to 8.0 m depth (Penuelas and Filella, 2003). The plant functional type (PFT) for this site is broadleaf evergreen tree (BET).

## 2.2 France, Puechabon [FR-Pue]

The Puechabon research site (43.4417° N-3.5944° E) is characterized by soil limitations linked to the hard Jurassic limestone formation beneath it (Cabon et al., 2018). The soil does not have a clear differentiation of horizons (Shahin et al., 2013), with a silty clay loam soil texture (Reichstein et al., 2002). The superficial soil layers have a high soil permeability thanks to the elevated stone fraction of 0.75. On the other hand, soil layers beneath 50 cm have a larger clay content (> 30 %) and a stone fraction of more than 0.9 (Limousin et al., 2009; Pita et al., 2013). This location has a Mediterranean climate with a mean annual precipitation of 1023.0 mm yr$^{-1}$ and a mean annual temperature of 13.8 °C (Poyatos et al., 2021). The forest canopy is dominated by *Quercus ilex*, while the understorey is sparse and dominated by shrubs such as *Buxus sempervirens* L., *Phyllirea latifolia* L., *Pistacia terebinthus L.* and *Juniperus oxycedrus* L. (Allard et al., 2008). This forest stand allocates most of the fine root biomass in the top 50 cm of the soil profile with a small fraction of roots reaching depths down to 4.5 m (Allard et al., 2008). Similarly to ES-Alt, the PFT is classified as BET, due to the dominance of *Quercus ilex*.

## 2.3 France, Hesse [FR-Hes]

The Hesse experimental site ($48.6742°$ N-$7.0647°$ E) is located on top of a sandstone formation, with a gentle southern slope (Le Goff and Ottorini, 2001a), and a soil classified as Luvisol/stagnic luvisol with a maximum soil depth of 145 cm depth (Granier et al., 2007). It is a clear transition between the eluviated horizon and the horizon with high clay accumulation at 50 cm depth (Zapater et al., 2012), leading to a low clay content in the upper soil layers (Granier et al., 2000b). It has a semi-continental climate with a mean annual temperature of $10.0\,°C$ and a mean annual precipitation of $1003\,mm\,yr^{-1}$ (Poyatos et al., 2021). The vegetation is dominated by *Fagus sylvatica* L. with 90 % of the trees, with the remaining 10 % represented by *Carpinus betulus* L., *Betula pendula* Roth, *Quercus petraea* (Matt.) Liebl., *Larix decidua* Mill., *Prunus avium* L., and *Fraxinus excelsior* L. (Granier et al., 2000a). Understorey vegetation is sparse as a consequence of a closed canopy and the mineral soil is covered by a mull type humus (Daniel Epron et al., 2004; Le Goff and Ottorini, 2001a). *Fagus sylvatica* trees allocate most of the fine roots in the upper 40 cm of soil with some fine roots reaching depths down to 1.5 m (Betsch et al., 2011; Granier et al., 2000b, a; Zapater et al., 2012). The dominant PFT is classified as broadleaf deciduous tree (BDT).

## 2.4 Russia, Fyodorovskoye [RU-Fyo]

The Fyodorovskoye experimental site ($56.4615°$ N-$32.9221°$ E) is located in the Central Forest Reserve in the Tver region, Russia. The soils are classified as Eutric Podzoluvisol and Gleyic Podzolavisol and characterized by their poor drainage, poor soil aeration, and bog growth on the surface (Vygodskaya et al., 2002). The site lithology describes a previous vast periglacial lake at 8 m depth, above which the peatland started forming (Novenko and Zuganova, 2010) and still present. The peat layer at this site has an average depth of 50 cm (Arneth et al., 2002), and the glacial deposits result in a loamy texture of the soil beneath it (Novenko and Zuganova, 2010; Schulze et al., 2002). The water table at this site is shallow, forcing the trees to allocate most of the fine roots in the top 20 cm of the soil (Milyukova et al., 2002). The climate is a transition between European Oceanic to continental climate being classified as moderately continental (Schulze et al., 2002) with a mean annual precipitation of $719.7\,mm\,yr^{-1}$ and a mean annual temperature of $4.7\,°C$ (Poyatos et al., 2021). The forest stand is dominated by the tree species *Picea abies* (L.) Karst (Norway spruce), *Betula pubescens* L. (Birch) and some *Pinus sylvestris* L. (Scots pine) (Kurbatova et al., 2013; van der Laan et al., 2014). The dominance of coniferous trees at this site determines the needleleaf evergreen tree (NET) PFT classification.

## 3 Methodology

### 3.1 Model Implementation

CLM 5 (Lawrence et al., 2019) was implemented at each experimental site using point-scale setups. Hourly atmospheric forcings (precipitation, wind speed, air temperature, relative humidity, atmospheric pressure, and incoming shortwave radiation) were retrieved from the SAPFLUXNET data set. Incoming longwave radiation was determined based on air temperature and actual air vapor pressure according to An et al. (2017), while missing variables (i.e, atmospheric pressure) and missing data were filled using COSMO-REA6 reanalysis product (Bollmeyer et al., 2015). The plant functional types (PFTs) broadleaf evergreen tree (BET), broadleaf deciduous tree (BDT), and needleleaf evergreen tree (NET) describe the vegetation cover of the different experimental sites. CLM 5 implements a default parameterization based on previous published data for the root distribution (Jackson et al., 1996) and plant hydraulics (Kattge et al., 2011; Kennedy et al., 2019), and the other plant physiological properties (e.g., aerodynamic and photosynthesis) according to the PFT classification (Lawrence et al., 2019) of each experimental site. Site-specific monthly leaf area index (LAI) values were computed based on the 1 km Global Land Surface Satellite (GLASS) product provided every 8 days (Liang et al., 2013, 2014). In order to account for inter-annual variability in the plant phenological development, yearly model runs were performed where the LAI information was updated at the start of each run. Default soil texture profiles (Bonan et al., 2002) and default depth to bedrock (Pelletier et al., 2016) were used at each site; see Table 1 for summary information. Although this data set does not reflect the exact site conditions, the depth to bedrock is closer to measurements reported in previous studies at the experimental sites. The simulations were carried out over different time periods (i.e, 2012-2014 at ES-Alt, 2001-2011 at FR-Pue, 2001-2005 at FR-Hes, and 2001-2003 at RU-Fyo) covering some of most extreme drought events in Europe (e.g., 2003 and 2006). Finally, soil moisture and soil temperature were initialized by performing multi-year spin-up runs, with CLM 5 repeatedly reinitialized until dynamic equilibrium condition was reached.

### 3.2 Bedrock configurations

In CLM 5 the plant access to soil water is controlled by the PFT-specific root distribution parameter ($\beta$) based on the formulation proposed by Jackson et al. (1996). This parameterization results in an exponentially decreasing root profile with soil depth, which is truncated by the position of the bedrock. The water acquisition by the plants is constrained by the effective soil depth as set by the DTB static parameter, which implies that all soil layers beneath DTB are impermeable and without water holding capacity; see schematic of Fig. 1b for a graphical illustration of the intersection between root distribution and DTB as conceptualized in CLM 5.

The first configuration (default model configuration: DMC) uses the default soil texture and DTB parameters for each location and it is used as a baseline for comparison with the other configurations. The second configuration (deeper bedrock configuration: DBC) shifts the bedrock depth of all sites down by 1.5 m, while keeping the same default soil texture classification. This

depth was defined as a fixed parameter across all sites because the thickness and degree of bedrock weathering are difficult to characterize over broad scales (Holbrook et al., 2014), where the interaction between climate, vegetation, and rock type determines the extent and properties of the weathered bedrock (Pawlik et al., 2016). The third configuration (fractured bedrock configuration: FBC) uses the extended DTB of the second configuration, but parameterizes the soil texture of the layers laying in between the original and the new DTB as 90 % sand and 10 % clay (Table 1). This approach aims to mimic the hydrological

behavior of a fractured bedrock based on two main assumptions: a fractured bedrock should have a high–water conductivity and low water holding capacity. As the sandy soil texture classification can be described by any soil with a combination of more than 85 % of sand and less than 10 % of clay, we decided to choose the combination of soil fractions that provides a sandy soil texture with a maximum water holding capacity for this textural class. The high sand percentage will mimic the fast water movement through the primary and secondary porosity of the fractured bedrock. At the same time, the low clay content allows

having a low water holding capacity for plant water uptake compared with the above soil layers.

### 3.3  Data Analysis

#### 3.3.1  Reference evaporation and observed transpiration

To assess the effect of water stress on transpiration rates, the unstressed reference crop evaporation ($E_o$) was calculated following Allen et al. (1998) using Equation 1. This equation assumes a reference crop of 0.12 m height characterized by a surface

resistance of $70\,\mathrm{s\,m^{-1}}$ and an albedo of 0.23. This equation requires wind speed ($u$) in $\mathrm{m\,s^{-1}}$, net radiation ($R_n$) and ground heat flux ($G$) both in $\mathrm{MJ\,m^{-2}d^{-1}}$, air temperature ($T$) in °C, the actual and saturated vapour pressures ($e_a$ and $e_s$, respectively) in kPa. $G$ was extracted from the modeled results of the DMC for each site. The slope of the saturation vapour pressure curve at air temperature ($\Delta$, kPa°C) was computed using equation 2. The psychrometric constant ($\gamma$) was estimated with equation 3, where $\lambda$ is the latent heat of vaporization ($2.45\,\mathrm{MJ\,kg^{-1}}$), $c_p$ is the specific heat at constant pressure ($1.013\times10^{-3}$

$\mathrm{MJ\,kg^{-1}\,°C^{-1}}$), $p$ is the atmospheric pressure (kPa), and $\epsilon$ is the molecular weight ratio of water vapour and dry air (0.622).

$$E_o = \frac{0.408 \cdot \Delta \cdot (R_n - G) + \gamma \frac{900}{T+273} \cdot u \cdot (e_s - e_a)}{\Delta + \gamma \cdot (1 + 0.34u)} \tag{1}$$

$$\Delta = \frac{4098 \cdot \left(0.6108 \cdot \exp\left(\frac{17.27 \cdot T}{T+237.3}\right)\right)}{(T + 237.3)^2} \tag{2}$$

$$\gamma = \frac{c_p p}{\epsilon \lambda} \tag{3}$$

Hourly and sub-hourly sap flux of individual trees ($Q_{\text{tree}}$) in cm$^3$ hr$^{-1}$ was retrieved from the SAPFLUXNET data set (Poyatos et al., 2021) for each experimental site and aggregated to daily fluxes (m$^3$ d$^{-1}$). Daily transpiration fluxes were upscaled to stand transpiration ($E_{\text{T}}$) in mm d$^{-1}$ using equation 4 and following the recommendations by Nelson et al. (2020). This equation requires the transpiration flux ($Q_{\text{tree}}$) in m$^3$d$^{-1}$tree$^{-1}$, the tree basal area ($\Upsilon_{\text{tree}}$) in m$^2$ tree$^{-1}$, the stand basal area ($\Upsilon_{\text{stand}}$) in m$^2$ m$^{-2}$, and the number of measured trees ($n$). All information required in Equation 4 was extracted from the SAPFLUXNET data set for each site.

$$E_{\text{T}} = \frac{\Upsilon_{\text{stand}}}{n \cdot 10^3} \cdot \sum_{\text{tree}=1}^{n} \frac{Q_{\text{tree}}}{\Upsilon_{\text{tree}}} \tag{4}$$

At each site the daily standard deviation of transpiration rates ($E_{\text{T}-\sigma}$) provides an indication of the different response of individual trees (including different species) to the environmental drivers. Table 1 shows the maximum ($\sigma_{\text{max}}$), median ($\sigma_{\text{median}}$), and mean ($\sigma_{\text{mean}}$) values of $E_{\text{T}-\sigma}$ for the selected time periods. FR-Hes is the site where the sampled trees have the largest variability in the transpiration response ($\sigma_{\text{mean}}$: 0.71 mm d$^{-1}$, $\sigma_{\text{median}}$: 0.62 mm d$^{-1}$), while the other three sites show similar median and mean values close to $\sim$0.3 mm d$^{-1}$. FR-Pue and FR-Hes display $\sigma_{\text{max}}$ values (3.53 mm d$^{-1}$ and 4.3 mm d$^{-1}$, respectively) that are three-fold larger than those in ES-Alt and RU-Fyo ($\sigma_{\text{max}}$: $\sim$1.0 mm d$^{-1}$). These large $\sigma_{\text{max}}$ suggest the strong intra-specific variability of transpiration response to the meteorological conditions for forest stands dominated by a single tree species. Sites such as ES-Alt and RU-Fyo characterized by mixed forest stands experience a more homogeneous transpiration response among the sampled trees.

### 3.3.2 Index of Agreement

The daily stand transpiration ($E_{\text{T}}$) in mm d$^{-1}$ was compared to the simulated transpiration of the three model configurations (see section 3.2). The relative comparison of the three bedrock representations was carried out by applying a symmetric index of agreement ($\Gamma$) proposed by Duveiller et al. (2016). The index is calculated using Equation 5 and is based on the product between the Pearson correlation coefficient ($r$) and an $\alpha$ coefficient, which scales $r$ as a measure of agreement. When $r$ is negative, $\alpha$ becomes zero under the consideration that a negative correlation does not show agreement when comparing against a benchmark. Consequently, $r$ is a measure of the linear agreement/dependence reflecting how well the measured and simulated transpiration time series agree in terms of their temporal deviations with respect to their mean responses, and the term

$\alpha$ represents any bias (additive/multiplicative) between the two data sets and ranges between 1 (no bias, perfect agreement) and 0 (full bias, no agreement).

$$\Gamma = \alpha \cdot r \text{ ; where } \alpha = \begin{cases} 0 & \text{, if } r \text{ is} \leqslant 0 \\ \dfrac{2}{\frac{\sigma_X}{\sigma_Y} + \frac{\sigma_Y}{\sigma_X} + \frac{(\overline{X}-\overline{Y})^2}{\sigma_X \cdot \sigma_Y}} & \text{, otherwise} \end{cases} \tag{5}$$

### 3.3.3 Plant Vulnerability Curve

The physiological implication of the plant response based on the three bedrock configurations is analyzed using the vulnerability curve of the plant hydraulic system as implemented in CLM 5 (Kennedy et al., 2019). Equation 6 expresses the plant segment hydraulic conductivity ($k$, $s^{-1}$) as a function of tissue water potential ($\psi$, MPa) and contains three parameters: the water potential at 50 % loss of conductivity ($\psi_{p50}$) in MPa, the maximum conductivity ($k_{max}$) of the plant segment in $s^{-1}$, and the non-dimensional sigmoidal shape-fitting parameter of the curve ($c_k$). These parameters are defined at the PFT-level and reported in Table 1 for the different selected sites.

$$k = k_{max} \cdot 2^{-\left(\frac{\psi}{\psi_{p50}}\right)^{c_k}} \tag{6}$$

The loss of hydraulic conductivity in a plant segment ($\Xi_{PLC}$) due to low tissue water potential can also be expressed as the percent of conductivity ($\Xi$) (Equation 7), with 0 % for complete loss of conductance, 50 % representing the conductance at $\psi = \psi_{p50}$ and 100 % representing no loss.

$$\Xi = \frac{k}{k_{max}} \cdot 100 \tag{7}$$

## 4 Results

### 4.1 Reference Evaporation and Transpiration Fluxes

The reference evaporation ($E_o$) approximates the canopy water demand under optimal soil water supply and its mean annual value can be compared to mean annual precipitation ($P$) to quantify the climatic water stress at a given site. For the selected study sites (Table 2), FR-Pue has the largest annual $E_o$ (921 mm yr$^{-1}$) and the largest annual $P$ (915 mm yr$^{-1}$), while RU-Fyo

has both the smallest annual $E_\mathrm{o}$ (480.3 mm yr$^{-1}$) and $P$ (405 mm yr$^{-1}$). ES-Alt and FR-Hes have similar values of annual
$E_\mathrm{o}$ (753.5 mm yr$^{-1}$ and 728.4 mm yr$^{-1}$, respectively), but very different mean annual $P$ (465 and 900 mm yr$^{-1}$, respectively),
indicating that in ES-Alt the ecosystem is on average water stressed while in FR-Hes is not. At FR-Pue and ES-Alt, the months
with the highest $E_\mathrm{o}$ are those with the lowest $P$, implying that at these sites the ecosystems are potentially subject to a strong
seasonal climatic drought. On the contrary, FR-Hes and RU-Fyo do not experience pronounced dry periods (Fig. 2). Note that
annual $E_\mathrm{o}$ is not balanced by the annual contribution of $P$ at almost all sites, with the exception of FR-Hes where $P$ exceeds
$E_\mathrm{o}$. This water deficit may lead important restrictions in soil moisture during dry periods at sites such as FR-Pue and ES-Alt.

The sites of ES-Alt, FR-Hes, and FR-Pue show a similar monthly trend for the maximum vapor pressure deficit (Fig. 2). ES-
Alt experiences an extended drier period (three months) while RU-Fyo is the only site where the maximum vapor pressure
deficit do never surpasses 3 kPa. Relationships between $E_\mathrm{T}$ and $E_\mathrm{o}$ (Fig. 2) illustrate the difference between atmospheric
water demand and plant transpiration across the selected sites, where the upscaled $E_\mathrm{T}$ represents the integrated effect of
environmental constraints (i.e, soil moisture availability and atmospheric demand) on the ecosystem response. RU-Fyo and
ES-Alt are the sites with the smallest annual transpiration rates of 81.5 mm yr$^{-1}$ and 177 mm yr$^{-1}$, respectively (Table 2).
The large atmospheric water demand in ES-Alt is not satisfied by the available soil moisture, which defines a soil water-
limited transpiration process. FR-Pue and FR-Hes have a similar annual $E_\mathrm{T}$ with values ranging between 329.7 mm yr$^{-1}$ and
350.1 mm yr$^{-1}$, respectively (Table 2).

Daily transpiration rates show differences in terms of timing and magnitude among sites (Fig. 3). FR-Hes has the highest $E_\mathrm{T}$
with a peak value of 4.5 mm d$^{-1}$ at the beginning of summer followed by FR-Pue with 2.0 mm d$^{-1}$ at the beginning of June.
$E_\mathrm{T}$ in FR-Hes shows a quick increment from 0 mm d$^{-1}$ at the beginning of April to more than 2 mm d$^{-1}$ one month later. This
increment is directly linked to the leaf flushing period of the dominant tree species (*F. sylvatica*). $E_\mathrm{T}$ declines consistently
throughout the summer at all sites, with a striking decline in FR-Pue where the average daily transpiration changes from
1.77 mm d$^{-1}$ to 0.72 mm d$^{-1}$ between June and July. This abrupt pattern shows how the transpiration process is constrained
by low soil moisture and large atmospheric water demand. Overall, the temporal analysis of $E_\mathrm{o}$ and $E_\mathrm{T}$ suggests a pairwise
clustering of FR-Hes and RU-Fyo as sites with a more homogeneous temporal distribution of $P$ that controls the atmospheric
humidity (Granier et al., 2008). In a similar way, ES-Alt and FR-Pue can be clustered together as soil water-limited sites
(Grossiord et al., 2015, 2018) where the large atmospheric moisture deficit is not satisfied by the soil water supply. Altogether,
$E_\mathrm{T}$ is smaller than $E_\mathrm{o}$ in ES-Alt, FR-Pue, and RU-Fyo, with a sudden decline of $E_\mathrm{T}$ at FR-Pue and ES-Alt in the middle of
the year. FR-Hes transpires almost at $E_\mathrm{o}$ in spring and summer. Finally, RU-Fyo transpiration rates are less than 50 % of $E_\mathrm{o}$
throughout the season, despite sufficient rainfall to satisfy annual evaporative demand.

## 4.2 Modelling Effects of Bedrock Configuration

Figure 3 shows the multi-annual variability of the measured and simulated daily transpiration fluxes at the selected sites. The visual inspection of these plots illustrates the different capability of CLM 5 to capture the intra- and inter-site variability of the measured transpiration fluxes. In ES-Alt and FR-Pue site, the DMC largely overestimates $E_T$ during spring and underestimates $E_T$ in summer; at both sites the model simulates a sharp decline in $E_T$ at the beginning of summer. The introduction of a deeper (i.e, DBC) and fractured (i.e., FBC) bedrock configuration alleviates the summer underestimation without eliminating the large overestimation during spring. In FR-Hes, the DMC configuration accurately reproduces the magnitude and intra-seasonal variability of the measured transpiration fluxes, with a slight overestimation of $E_T$ during early spring and summer. The modified bedrock configurations DBC and FBC slightly increase the summer overestimation of $E_T$ with respect to the default configuration. All three model configurations (i.e., DMC, DBC, and FBC) systematically overestimate $E_T$ in RU-Fyo, with no differences among model configurations.

The soil parameterization used in DMC and DBC agrees with the published data for FR-Hes, FR-Pue, and RU-Fyo (Figure A2). These configurations are located within the boundaries of the soil texture classification of each site. In this regard, the soil water storage capacity, and the infiltration and percolation rates are expected to be representative of the expected site conditions. In ES-Alt, the clay content used in the model is similar with the site conditions, providing similar water holding capacity. However, the sand content used in the DMC and DBC model configurations reflects a lower hydraulic conductivity than the one expected for this site.

A quantitative estimation of the performances of the three model configurations is obtained using the index of agreement described in section 3.3.2. Figure 4 shows the monthly variability of the Pearson correlation coefficient (r), $\Gamma$ index, and $\alpha$ coefficient for the three model configurations at the four experimental sites. In ES-Alt, the DMC configuration reproduces the measured transpiration with an $r$ coefficient systematically larger than 0.6 and it peaks around 0.8 for the period between November and May (Table A2). This temporal agreement is, however, concurrent with a large bias, $\alpha$ coefficient ranging between 0.24 and 0.94, which determines $\Gamma$ values being not larger that 0.57. The performance of the DMC configuration drastically deteriorates during summer and fall (max $\Gamma$ equals to 0.29), where negatively or poorly correlated time series in June, September, and October determine zero or close to zero values of $\Gamma$. Note that during the dry season decent correlation values (i.e, August) are greatly outweighed by the bias between measured and simulated transpiration values (Fig. 4).

The performance of the DMC configuration shows less temporal variability at the FR-Pue study site. With the exception of a few months (e.g., December), the correlation ($r$) and the $\alpha$ index vary between 0.41 and 0.79 and between 0.45 and 0.96, respectively, with the resulting $\Gamma$ values between 0.25 and 0.69 (Table A2). Although not as strongly as at ES-Alt, the overall performance of the DMC configuration tends to decrease during the dry period (June-September) also at FR-Pue. The performance of the DMC is striking at FR-Hes during the May-October period (Fig. 4), with $r$ coefficient values larger than 0.65, $\alpha$ values larger than 0.85, and resulting $\Gamma$ index between 0.58 and 0.86. On the other hand, the DMC performance at FR-Hes is

negatively affected by the strong seasonal pattern of vegetation phenology, as the leaf shedding period (November), leaf-less period (December to March), and leaf flushing period (April) have the lowest $\Gamma$ values (0.62, 0.0, and 0.23, respectively). This discrepancy is likely caused by the leaf area index (LAI) used in the model, with values between $0.9\,\mathrm{m^2\,m^{-2}}$ and $1.5\,\mathrm{m^2\,m^{-2}}$ during the leaf-less period. This results in simulated transpiration when trees on site do not have leaves at all. RU-Fyo is the only experimental site where the capability of the DMC configuration in reproducing the transpiration response is systematically low throughout the year. At this site, the satisfactory performances in terms of temporal correlation (i.e., $r$) in April (0.71) and June (0.65) are accompanied by a high bias, which reduces the $\Gamma$ values (i.e., 0.47 and 0.27, respectively). Overall, the DMC configuration shows systematic model deficiency in reproducing the dry-season (i.e., June-September) transpiration response at Mediterranean sites with relatively high atmospheric water demand and shallow (below 1 m) bedrock depth (i.e., Es-Alt and FR-Pue).

Figure 4 shows the impact of adopting different bedrock configurations on the performance skills of the model across the four experimental sites. The visual inspection of the plot reveals that neither of the DBC and FBC configurations alter the model performance during November-May at any of the selected sites. Interestingly, neither the temporal agreement nor the bias are modified by the extended and altered bedrock during this period. At the RU-Fyo site, these two alternative bedrock configurations do not affect the model estimates at all. On the other hand, the two proposed configurations (i.e., DBC and FBC) have a clear impact on the simulated transpiration at ES-Alt and FR-Pue between June and September (Fig. 4). At ES-Alt, both configurations improve the $r$ coefficient and the $\Gamma$ index, improving the model's capability to realistically simulate transpiration fluxes. Such improvement is most evident in the late summer (i.e., August and September) and to a lesser extent in October (Fig. 3). In particular, the DBC configuration shows slightly larger $\Gamma$ values in September and October, while FBC improves the modeled transpiration response in July and August (Table A2). The direction of changes introduced by DBC and FBC configurations are less distinct in FR-Pue site. Here, extending the bedrock depth (i.e., FBC configuration) deteriorates the model performance while prescribing a different water holding capacity in the permeable bedrock (i.e., DBC) improves the model response just in August, September, and October. In particular, the improvements obtained using the DBC configuration are mostly explained by better temporal correlation values and marginal changes in model bias. Finally, both modified bedrock configurations at the FR-Hes site decrease the overall fit of the simulated transpiration response. This is particularly clear in August where DBC and FBC configurations drastically decrease the temporal agreement and overestimate the daily cumulative values compared to the estimates of the default (i.e., DMC) model configuration and the measured transpiration values (Fig. 3).

## 4.3 Plant Response to Water Stress

Figure 5 illustrates the percent of hydraulic conductance ($\Xi$) as a function of plant tissue water potential and the distribution of simulated $\Xi$ values during the dry period of the year (i.e., June-October). The shape of the plant vulnerability curves from where figure 5 is based on is highly dependent on $\psi_{\mathrm{p50}}$ (see Equation 6) and the distribution of values indicates the level of

water stress at which plants are operating (See Fig. A1 to Fig. A6). For instance, under well-watered conditions the plant is unstressed and the experienced water potential is close to 0, allowing the corresponding plant organ (i.e., xylem and sunlit leaves) to move water at their maximum capacity. As soil water uptake becomes limiting, plant water potential decreases, followed by the xylem (or leaf) conductance, and the vegetation starts experiencing water stress conditions.

The most extreme plant water stress conditions are simulated at ES-Alt, where the model simulates a reduction in the median conductivities to 20 % (xylem) and 10 % (leaves) in August and even below 10 % in September. Extreme plant stress conditions are also simulated by the DMC configuration at FR-Pue experimental sites. At this location, the xylem conductivities are above 80 % of their maximum value for more than half of the simulated time in July, whereas their median value drastically decreases below 30 % in August. The range of water potentials is even more severe at the leaf level, which simulates a drop in the median conductivity from 45 % to 10 % between July and August. The system smoothly recovers in September with medians of 55 % and 30 % of the maximum conductivity for xylem and leaf, respectively. At the FR-Hes site, the DMC configuration results in little to moderate water stress conditions (majority of simulated data is well above $\psi_{\mathrm{p50}}$, see Fig. 5), with some outliers due to isolated episodes of extreme heat (e.g., August-2003) and/or dry soil moisture conditions (e.g., July-2003). Finally, the analysis of the $\Xi$ at RU-Fyo suggests well-watered soil conditions throughout the year, with xylem and leaf conductivity larger than 95 % of their maximum values.

The effect of deeper or fractured bedrock configurations on simulated plant water status emerges most clearly wherever the default configuration results in severe water stress, i.e. mainly at the Mediterranean sites, ES-Alt and FR-Pue (Fig. 5). At ES-Alt, the DBC shifts the median xylem conductivity to 60-80 % of their maximum, compared to 10-20 % in the default configuration in August-September. The effect of the FBC configuration is similar, but less pronounced at this site. At FR-Pue, the DBC configuration leads to a very strong reduction of water stress compared to the default configuration, with increases of relative xylem conductivity from 15-50% to 90-100% in August, and even higher in the other months. The shift is even more clear when inspecting the loss of conductivity simulated at the leaf level. In this case, there is also a drastic change in the inter-quartile range, especially in July and September, suggesting a reduction of the inter-annual variability of vegetation response during the dry season. Yet, the implementation of the FBC at FR-Pue alleviates the harsh conditions simulated by DMC, but the leaves are still affected by a severe plant water status conditions. On the other hand, at FR-Hes and RU-Fyo, where DMC results in little loss of conductance, both the DBC and FBC model do not have any clear effects on the simulated plant water status, except for August and September at FR-Hes, where the rare excursions of relative hydraulic conductivity below the 80% mark are removed.

## 5 Discussion

Plants rely on structural (e.g., rooting depth, leaf thickness) and functional (e.g., stomatal regulation, plant storage capacitance, hydraulic redistribution) strategies to tolerate extended dry periods (Aroca, 2012; Gupta et al., 2020). These strategies depend

on the upper (i.e., climate) and lower (i.e., soil and geological) boundary conditions (Fan et al., 2017). If geological conditions allow for the formation of deep soils (e.g., Amazon Basin, Loess Plateau), the roots access deep groundwater which is very

important for surviving extended dry periods (Chitra-Tarak et al., 2021; Tao et al., 2021). On the other hand, when soils are shallow and less developed, the trees must thrive by accessing additional water pockets in the weathered bedrock. The heterogeneous nature of weathered bedrock depends on the interaction between climate, vegetation, and rock type (Pawlik et al., 2016). These interactions allow the increment of the water-holding capacity of weathered bedrock by increasing the porosity and mineral surface area (Navarre-Sitchler et al., 2015). This water holding capacity is considered negligible (Novák

and Surda, 2010), but the vertical extent of this layer makes the water reservoir large enough to support deep rooting vegetation during dry spells (Graham et al., 2010; Jones and Graham, 1993). As an example, Mediterranean trees are able to uptake water from the deep vadose zone (Carrière et al., 2020) sustaining transpiration during the dry season without being affected by embolism (David et al., 2007, 2013; Prieto and Ryel, 2013). Given the strong effect of transpiration on the land surface energy partitioning (Duveiller et al., 2018; Forzieri et al., 2020), it is important that LSMs correctly represent plant water uptake

processes and their link to the magnitude and timing of transpiration. The advanced plant hydraulics representation of CLM 5 simulates the water uptake and transport across the whole plant system (e.g., roots, stems, and leaves). This allows to evaluate the influence of bedrock configuration on the bulk transpiration flux as well as to differentiate its impact on the susceptibility to hydraulic failure of the different plant segments.

## 5.1 Bedrock Effects on Modeled Transpiration

The rooting profile implemented in CLM 5 follows an exponential distribution, resulting in non-zero root abundance throughout all the way down to the bedrock. The PFTs BDT and BET allocate 95 % of the roots within the first 0.7 m of soil. Those sites with an impermeable bedrock layer very close to the surface effectively cut off the root distribution at that position, without increasing root abundance in the layers above. This limits drastically the access to water resources forcing the model to use only the water available in the superficial soil layers, affecting transpiration rates and plant hydraulic response under dry conditions.

As a result, the DMC configuration restricts the water acquisition of these PFTs at seasonally dry sites, such as ES-Alt and FR-Pue, by removing almost 10% of the roots that can provide more than 50 % of transpiration water during summer conditions in dry environments with deeper soils (Carrière et al., 2020; Klos et al., 2018; Hahm et al., 2020; Jiang et al., 2020; Nardini et al., 2021; Qi et al., 2018; Rempe and Dietrich, 2018). Oak tree species are known to access deep water storage because of their extensive rooting depths (Gil-Pelegrín et al., 2017). The *Quercus ilex* trees growing at ES-Alt and FR-Pue have shown

this feature (Baldocchi et al., 2010; Forner et al., 2018) allowing the trees to transpire during the dry season despite the low soil water potentials. The transpiration signal retrieved from sap flow sensors shows how plants are able to access deep water sources during dry periods when compared to $E_T$ as simulated by the DMC configuration. The steep temporal decline of modeled $E_T$ in ES-Alt and FR-Pue during summer contrast significantly with the measured values at both sites (Fig. 3). These differences depict the dry out process of the superficial soil layers carried out by the vegetation due to the lack of access to

deep water sources by the modeled vegetation and the absence of $P$ to replenish the transpired soil water. *Fagus sylvatica* trees

strongly rely on shallow soil water because of their superficial root system (Lüttschwager and Jochheim, 2020), a condition documented at FR-Hes by Granier et al. (2000a) and Zapater et al. (2012). Meanwhile, during drought periods, this tree species uses the water stored in the trunk and roots as a reservoir to maintain transpiration until the next rainfall event (Betsch et al., 2011). However, the limited access of vegetation to deep water sources due to the shallow DTB is counterbalanced by the temporal distribution of $P$ that replenishes the transpired superficial soil water avoiding plant water stress conditions at this site.

The DTB parameterization used in CLM 5 simulations corresponds to a sharp transition between soil and consolidated bedrock assuming this latter as a hydrologically inactive layer (Lawrence et al., 2019) that limits the plant water supply at sites with thin soils and pronounced dry seasons. This assumption neglects the fact that weathered bedrock contains cracks formed by physical and chemical weathering processes (Pawlik et al., 2016; Phillips et al., 2019; Pope, 2015), allowing for accumulation of sediments and increasing its capacity to store water. McCormick et al. (2021) underlined the importance of water stored in the weathered bedrock to fulfill the vegetation physiological needs during the growing season. Similarly, Pelletier et al. (2016) stressed the beneficial effect of adding this intermediate layer in LSMs as a reservoir for plants during dry periods. The high heterogeneity of the underlying subsurface characteristics is critical for quantifying the water budget at the local scale (Blyth et al., 2021) and it determines the large uncertainty on the estimation of the weathered bedrock depth. In our study, moving the DTB 1.5 m below the initial value allowed the vegetation to access a larger soil water storage. This configuration doubled the summer transpiration flux of the vegetation at ES-Alt and FR-Pue Mediterranean sites and increased of almost 25 % in FR-Hes with respect to the DMC configuration (Table 2).

The large spatial heterogeneity of bedrock saturated conductivity (Welch and Allen, 2014) and the fact that this property depends on the parental material (Huggett, 2007; Summerfield, 1991) increases the difficulty of incorporating this additional bedrock layer in LSMs. As an attempt to reduce this complexity, the FBC configuration proposed in our study modifies the additional water storage added in DBC by changing its soil texture. The clay fraction of DBC is larger than 10 %, condition that enhances the soil water storage with respect to FBC which has a larger saturated conductivity than DBC. The assumption of representing a fractured bedrock as a sandy soil is supported by the similarities in saturated hydraulic conductivity ($K_s$) that both substrata have. Sandy soils have a $K_s$ of $10^{-4}\,\mathrm{m\,s^{-1}}$ (Miyazaki, 1996; Pachepsky and Park, 2015), values that are within the range reported for weathered granite rocks that oscillates between $10^{-2}\,\mathrm{m\,s^{-1}}$ to $10^{-5}\,\mathrm{m\,s^{-1}}$ (Rouxel et al., 2010; Katsura et al., 2009). The FBC reduces the summer transpiration with respect to DBC in ES-Alt, FR-Pue, and FR-Hes of 18 %, 24 %, and 4 %, respectively (Table 2). This reduction of $E_T$ is the consequence of reducing the soil water storage by changing the clay content of the fractured bedrock layer in FBC (Fig. 1). As an example, when switching from DBC to FBC configuration, the clay content in ES-Alt and FR-Pue is changed respectively from 20 % and 40 % to a fixed value of 10 %. It is important to underline that the inclusion of DBC and FBC leads to no differences in the model transpiration response during winter and spring periods but allow to increase the summer $E_T$ in ES-Alt, FR-Pue, and FR-Hes with respect to DMC.

## 5.2 Bedrock and Hydraulic Plant Recovery

The implementation of plant hydraulics in terrestrial modelling allows to link the plant transpiration to soil water availability and vapor pressure deficit (Bonan et al., 2014; Liu et al., 2020). The photosynthesis drives the plant water needs and forces the plant hydraulic system to extract soil water at a maximum rate when soil water resources are unlimited. CLM 5 uses Equation 6 to represent the plant water supply of different plant organs as a function of gradients in water potentials. In this formulation, $\psi_{p50}$ modulates the plant water uptake according to the atmospheric water deficit, plant organ conductivity, and soil matric potential (Lawrence et al., 2019). The segmented application of Equation 6 per soil layer to simulate the root water uptake allows the modeled vegetation to switch the water uptake from dry to wet soil layers (Kennedy et al., 2019). This plant hydraulic formulation provides also the opportunity to diagnose the hydraulic function of the different plant organs (i.e., roots, stems, and leaves) and their susceptibility to hydraulic failure (McCulloh et al., 2019; Meinzer et al., 2009) as part of the model response to environmental stresses.

The three model configurations (i.e., DMC, DBC, and FBC) have a tendency to overestimate $E_T$ during spring in ES-Alt, FR-Pue, and FR-Hes (Fig. 3). This tendency reflects the model transpiration response under unlimited soil water conditions and increasing VPD (Fig. 2 and Fig. A1). Under such conditions the plant hydraulic system extracts soil water to satisfy the atmospheric water demand with the positive feedback of $P$ replenishing the transpired soil water. However, at sites with a superficial DTB and extended periods with a lack of $P$, this mechanism leads to a complete dry-out of the soil profile. The extreme soil water depletion pushes the plant matric potentials well below the $\psi_{p50}$ values prescribed for each PFT, generating physiological conditions at which plants hardly survive. As an example, the matric potentials simulated for stems and leaves at FR-Pue and ES-Alt are beyond the safety margins reported for broadleaf tree species (-0.9 MPa and -0.8 MPa for root-to-stem and stem-to-leaf, respectively) (Johnson et al., 2012). The inclusion of a deeper and fractured bedrock (i.e., DBC and FBC) alleviates this issue allowing the model to reproduce more realistic plant water potentials and transpiration rates during dry periods at FR-Pue and ES-Alt. The alternative bedrock configurations reduce also the susceptibility to hydraulic failure according to the experienced matric potentials by stems and leaves (Fig. 5). However, the extreme water stress in summer time in the DMC configuration could have resulted from an overestimation of water use during the early season, leading to depleted water reservoirs in summer. Although the DBC and FBC configurations improved the simulated transpiration and water stress experienced by the stem and leaves during summer by increasing the soil water availability, neither of the configurations reduced the transpiration in spring. This could suggest that the reason for the underestimated summer transpiration might not be an underestimated water storage capacity but some other model deficiency that is responsible for both, overestimated transpiration in spring and subsequently underestimated transpiration in summer.

The current plant hydraulic formulation of CLM 5 assumes a full recovery of the plant also when its organs experience water potentials below $\psi_{p50}$; FR-Pue and ES-Alt sites are clear examples of the implications of this assumption with $\Xi$ continuously ranging between 10 % and 100 % at the stem and leaf level in summer (Fig. A3 and Fig. A4). Full recovery after partial failure

of the hydraulic system may be possible when plants are adapted to such conditions (Sperry and Love, 2015) despite that after long dry periods the root-to-soil interface becomes a constraint to plant water uptake (Zarebanadkouki et al., 2016) before the soil gets moist again. Some plant species may have strategies to repair embolism damage in their tissues shifting the hydraulic efficiency-safety trade-off (Johnson et al., 2012; Klein et al., 2018), but it can not be considered as a norm across all vegetation types. In addition, plants depending on rock moisture can develop special water access strategies such as dynamic root systems

or mycorrhizae growing along the rock cracks and accessing water stored in or dripping from the bedrock (Schwinning, 2020). Experimental evidence documented in several studies (Johnson et al., 2018; Klein et al., 2018; Ocheltree et al., 2020; Rehschuh et al., 2020) indicates that plants tissues are affected when their water potentials go below $\psi_{\mathrm{p50}}$. As a result, the conducting tissues of most of the plant species are not able to recover the pre-drought hydraulic conductivity or experience embolism. This is an important omission in the current plant hydraulic system of CLM 5 that requires further model developments including

for instance plant mortality and a hydraulic conductivity recovery following a different vulnerability path.

The uncertainty in the parameterization of $\psi_{\mathrm{p50}}$ in the plant hydraulic system of CLM 5 can lead to large and systematic errors in the transpiration fluxes. In the RU-Fyo site, the three model configurations overestimate the measured $E_{\mathrm{T}}$ with the same order of magnitude (Table 2) and the same temporal pattern (Fig. 3). This is due to the default $\psi_{\mathrm{p50}}$ value of -5.2 MPa (Table 1) assigned to the NET PFT, which differs from the reported values of the tree species monitored at RU-Fyo (i.e., *Betula sp.*,

*P. abies*, and *P. sylvestris*) which have $\psi_{\mathrm{p50}}$ mean values of -3.7 $\pm \dot{0}.3$ MPa, -1.5 $\pm$ 0.6 MPa, and -3.1 $\pm \dot{0}.5$ MPa (Choat et al., 2012). Larger values of $\psi_{\mathrm{p50}}$ allow the modeled vegetation to extract more soil moisture to fulfill the atmospheric water demand increasing considerably the simulated $E_{\mathrm{t}}$ with respect to the measured one. Finally, the growth of a bog as the main ground cover beneath the canopy in RU-Fyo is an indication of the poor drainage of the site (Vygodskaya et al., 2002). Since it is a poorly drained peatland site, it may be that root water uptake is hampered by water logging and anoxia in the root zone for

extended periods of time, and consequently reducing tree transpiration (Angstmann et al., 2013).

### 5.3 Breaking the bedrock to release the roots

The occurrence and severity of extreme weather conditions like droughts and heat waves (He et al., 2020) highlight the importance of representing the vegetation's mechanisms to avoid or cope with the adverse effects of this new reality. The inclusion of rock moisture stored in the weathered bedrock may become important for correctly simulating plant transpiration during

dry periods (Rempe and Dietrich, 2018). The challenge to quantify this rock moisture is linked with the large uncertainty in determining the physical characteristics of this weathered bedrock (Pelletier et al., 2016) and the lack of spatially distributed field information related to the water storage properties of the weathered bedrock. Previous studies have highlighted the importance of groundwater and lateral flow for an improved simulation of transpiration fluxes in LSMs (Maxwell and Condon, 2016; Zeng et al., 2018). In contrast, other studies highlighted the importance of an extended rooting system with the same aim

(Fan et al., 2017; Ichii et al., 2009). However, the focus of LSMs on soil water neglects the interaction between these essential components and the weathered bedrock. In this work, we show that allowing the root system to access water stored in the

weathered bedrock improves the transpiration and plant water stress estimates during dry periods. The modeling community should address these two issues in a unified approach, eventually improving the water supply at sites with shallow soils and dry conditions. This unified approach, where we allow to break the bedrock and release the root profile, will create a new water reservoir that will refine the vulnerability assessment of forest ecosystems growing in regions with a tendency to experience drier conditions.

## 6 Conclusions

Experimental studies have demonstrated that bedrock-vegetation interactions involve a significant and vital water resource for plants during the dry season which is largely omitted in hydrological and land surface models. This study tested the impact of this omission in a state-of-the art LSM, CLM 5, by comparing the simulated transpiration response of three different bedrock configurations (i.e., default, deeper, and fractured bedrock). The two additional configurations mimic the effect of a deeper impermeable layer as well as the impact of an overlying weathered material on the impermeable bedrock. This overlying weathered bedrock is parameterized assigning a low clay content to account for its water holding capacity while the larger sand content enables quick drainage of water percolating from the soil. The three model configurations were tested at four forested sites included in the SAPFLUXNET measurement network and characterized by a shallow bedrock and contrasting atmospheric water demand.

The results of this study suggest that the presence of a shallow bedrock defined as a hydrological inactive layer (i.e., default model configuration) leads to strongly reduced water availability to plants during prolonged dry seasons and unrealistic water stress in root, xylem, and leaf tissues. The results show also the positive impact, especially at Mediterranean climate sites, of increasing the depth to bedrock and adjusting the clay content to mimic a weathered bedrock; the simulated transpiration significantly increases attaining a better match with the measured seasonal transpiration patterns. Moreover, these two additional configurations (i.e., deeper and fractured bedrock) reduce the water stress experienced by the modeled xylem and leaf plant segments keeping the percentage loss of tissue conductivity above 50 %, which is more in line with the range of water potentials at which plants operate. The model default configuration at sites with climates without prolonged dry periods has an optimal performance because the soil water sources are not fully depleted by the ecosystem needs. Moreover, sites with colder temperatures and more humid conditions are not affected by the implemented model configurations. Overall, the proposed weathered bedrock formulation allows the modeled vegetation to make full use of the root profile reducing the negative effect of the large soil control on transpiration when superficial soil layers have been dry out.

This work is a first attempt to understand how the bedrock parameterization of CLM 5 impacts the transpiration and provides some important insights on the sensitivity of the newly developed plant hydraulics system. This scientific issue has been explored by accounting for the uncertainty in the definition of the depth to the bedrock and by altering the texture composition of the soil overlying the impermeable depth. As a first order estimation, this approach provided an assessment on the impact of the

additional soil water volume and water holding capacity on the simulated plant conditions. However, as also advocated in previous studies, novel and more advanced parameterizations should also include the physical characterization (e.g., bulk hydraulic conductivity, tortuosity, porosity) of weathered bedrock to represent water movement and storage in a fractured porous media. This will allow for an improved representation of water and nutrient uptake from the soil and weathered bedrock column. Therefore, future studies should focus on identifying the most suitable weathered bedrock representation to be integrated into LSMs as well as to develop novel measurement techniques and strategies for model parameterizations such as rock moisture and water holding capacity at the large scale.

## 7   Code availability

The R scripts used for pre-processing, post processing, and plotting the information are available in ZENODO repository at the following link: https://doi.org/10.5281/zenodo.5153161 (Jiménez-Rodríguez, 2021).

## 8   Data availability

Sap flow data is available from the ZENODO repository at the following link: https://doi.org/10.5281/zenodo.3971689 (last access: 20 August 2021) (Poyatos et al., 2020). COSMO-REA 6 data can be downloaded from the opendata-FTP server at DWD at the following link: https://opendata.dwd.de/climate_environment/REA/COSMO_REA6/ (last access: 1 June 2021) (HErZ and DWD, 2020).

*Author contributions.* César Dionisio Jiménez-Rodríguez (CJR) and Mauro Sulis (MS) designed the numerical experiments. CJR carried out the numerical simulations and led the analysis with input from MS and Stanislaus Schymanki (SS). CJR wrote the manuscript with significant contributions from MS and SS.

*Competing interests.* The authors declare that they have no conflict of interest.

*Acknowledgements.* This work is supported by the Luxembourg National Research Fund (FNR) CORE programme (C19/SR/13652816/CAPACITY).

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

**Table 1.** Model parameters used by the three model configurations at the four experimental sites, including the standard deviation variability of daily transpiration estimates ($E_{T-\sigma}$) computed for each site.

| | | | ES-Alt | FR-Pue | FR-Hes | RU-Fyo |
|---|---|---|---|---|---|---|
| **DMC** | Plant Functional Type | | BET | BET | BDT | NET |
| | Root Distribution Parameter | $\beta$ | 0.966 | 0.966 | 0.966 | 0.976 |
| | Water potential at 50% loss of conductivity | p50 | -2.648 MPa | -2.648 MPa | -2.648 MPa | -5.197 MPa |
| | Maximum stem conductivity | $k_{max}$ | $2\,\mathrm{m\,s^{-1}}$ | $2\,\mathrm{m\,s^{-1}}$ | $2\,\mathrm{m\,s^{-1}}$ | $2\,\mathrm{m\,s^{-1}}$ |
| | Shape fitting parameter for | $c_k$ | 3.95 | 3.95 | 3.95 | 3.95 |
| | Soil Texture | | Loam\|Clay-Loam | Silty-Clay-Loam | Loam\|Clay-Loam | Loam |
| | Bedrock Depth | $z_{brck}$ | 0.821 m | 0.945 m | 1.016 m | 1.32 m |
| **DBC** | Soil Texture (New Soil Layers) | | Loam | Silty-Clay-Loam | Loam | Sandy-Loam |
| | Bedrock Depth | $z_{brck}$ | 2.321 m | 2.445 m | 2.516 m | 2.82 m |
| **FBC** | Soil Texture (Mimicked Fractured Bedrock) | | Sandy | Sandy | Sandy | Sandy |
| | Bedrock Depth | $z_{brck}$ | 2.321 m | 2.445 m | 2.516 m | 2.82 m |
| **$E_{T-\sigma}$** | Maximum Daily Standard Deviation | $\sigma_{max}$ | $1.06\,\mathrm{mm\,d^{-1}}$ | $3.53\,\mathrm{mm\,d^{-1}}$ | $4.30\,\mathrm{mm\,d^{-1}}$ | $0.94\,\mathrm{mm\,d^{-1}}$ |
| | Mean Daily Standard Deviation | $\sigma_{mean}$ | $0.42\,\mathrm{mm\,d^{-1}}$ | $0.34\,\mathrm{mm\,d^{-1}}$ | $0.71\,\mathrm{mm\,d^{-1}}$ | $0.30\,\mathrm{mm\,d^{-1}}$ |
| | Median Daily Standard Deviation | $\sigma_{median}$ | $0.35\,\mathrm{mm\,d^{-1}}$ | $0.29\,\mathrm{mm\,d^{-1}}$ | $0.62\,\mathrm{mm\,d^{-1}}$ | $0.26\,\mathrm{mm\,d^{-1}}$ |

**Table 2.** Mean annual estimates of potential evaporation ($E_o$), stand transpiration ($E_T$), and precipitation ($P$) per study site. Mean accumulated values of the modeled transpiration for the period under analysis (July to September) for the three bedrock configurations (DMC, DBC, and FBC).

| | | ES-Alt | FR-Pue | FR-Hes | RU-Fyo |
|---|---|---|---|---|---|
| $E_o$ | [mm yr$^{-1}$] | $753.5 \pm 62.9$ | $921.2 \pm 43.1$ | $728.4 \pm 99.5$ | $480.3 \pm 88.1$ |
| $E_T$ | [mm yr$^{-1}$] | $177 \pm 10.2$ | $329.7 \pm 39.7$ | $350.1 \pm 68.2$ | $81.5 \pm 29.3$ |
| $P$ | [mm yr$^{-1}$] | $465.4 \pm 99.7$ | $914.7 \pm 228.3$ | $900.3 \pm 231.6$ | $404.6 \pm 45.0$ |
| | | | | | |
| | | Period: July to September | | | |
| $P$ | [mm] | $41.4 \pm 13.3$ | $147.9 \pm 56.8$ | $244.6 \pm 79.4$ | $148.6 \pm 63.7$ |
| $E_o$ | [mm] | $380.8 \pm 18.1$ | $396.0 \pm 26.8$ | $316.0 \pm 66.1$ | $240.8 \pm 48.2$ |
| $E_T$ | [mm] | $59.7 \pm 17.6$ | $82.3 \pm 18.9$ | $170.5 \pm 44.3$ | $37.9 \pm 20.6$ |
| $E_{T-DMC}$ | [mm] | $37.8 \pm 30.2$ | $74.1 \pm 26.6$ | $188.1 \pm 35.5$ | $82.7 \pm 3.0$ |
| $E_{T-DBC}$ | [mm] | $88.4 \pm 39.7$ | $171.3 \pm 35.0$ | $231.9 \pm 17.6$ | $82.7 \pm 3.0$ |
| $E_{T-FBC}$ | [mm] | $72.9 \pm 46.8$ | $129.4 \pm 42.9$ | $222.5 \pm 11.6$ | $82.7 \pm 3.0$ |
| Years | | 3 | 10 | 5 | 3 |

**Table A1.** Summary of the main sub-surface characteristics of each experimental site.

| Main Site Characteristics | ES-Alt | FR-Pue | FR-Hes | RU-Fyo |
|---|---|---|---|---|
| Rooting Depth (m) | 8.0 | 4.5 | > 1.5 | 0.2 |
| Peat Layer Depth (cm) | None | None | None | 50 |
| Soil Depth (cm) | 20-40 \| 100 | 50 | 145 | N.A. |
| Soil Texture: % Clay | 25.9 | 39 | 25 | N.A. |
| Soil Texture: % Sand | 57.3 | 26 | N.A. | N.A. |
| Soil Texture: % Silt | 16.8 | 35 | N.A. | N.A. |
| Soil Type | Sandy Clay Loam | Silty Clay Loam | Clay Loam | Loam |
| Soil Permeability | N.A. | High | N.A. | N.A. |
| Superficial Stone Fraction | N.A. | 0.75 (0-50cm) | N.A. | N.A. |
| Deep Stone Fraction | N.A. | 0.90 (>50 cm) | N.A. | N.A. |
| Bedrock Type | Cretaceous carbonate | Jurassic Limestone | Sandstone | Glacial deposits |
| Water Table Depth | Deep | Deep | Deep | Superficial |
| Plant Water | Soil Water, Weathered Bedrock, Groundwater | Soil Water, Weathered Bedrock, Groundwater | Soil Water | Non-Saturated Substratum |
| Hydraulic Lift | Present | N.A. | Present | N.A. |
| References | Forner et al. (2018); Grossiord et al. (2015); Duque et al. (2008); Penuelas and Filella (2003) | Allard et al. (2008); Pita et al. (2013); Limousin et al. (2009) | Betsch et al. (2011); Granier et al. (2000b, a, 2007); Le Goff and Ottorini (2001b); Zapater et al. (2012) | Arneth et al. (2002); Kurbatova et al. (2002); Milyukova et al. (2002); Novenko and Zuganova (2010); Schulze et al. (2002); Vygodskaya et al. (2002) |

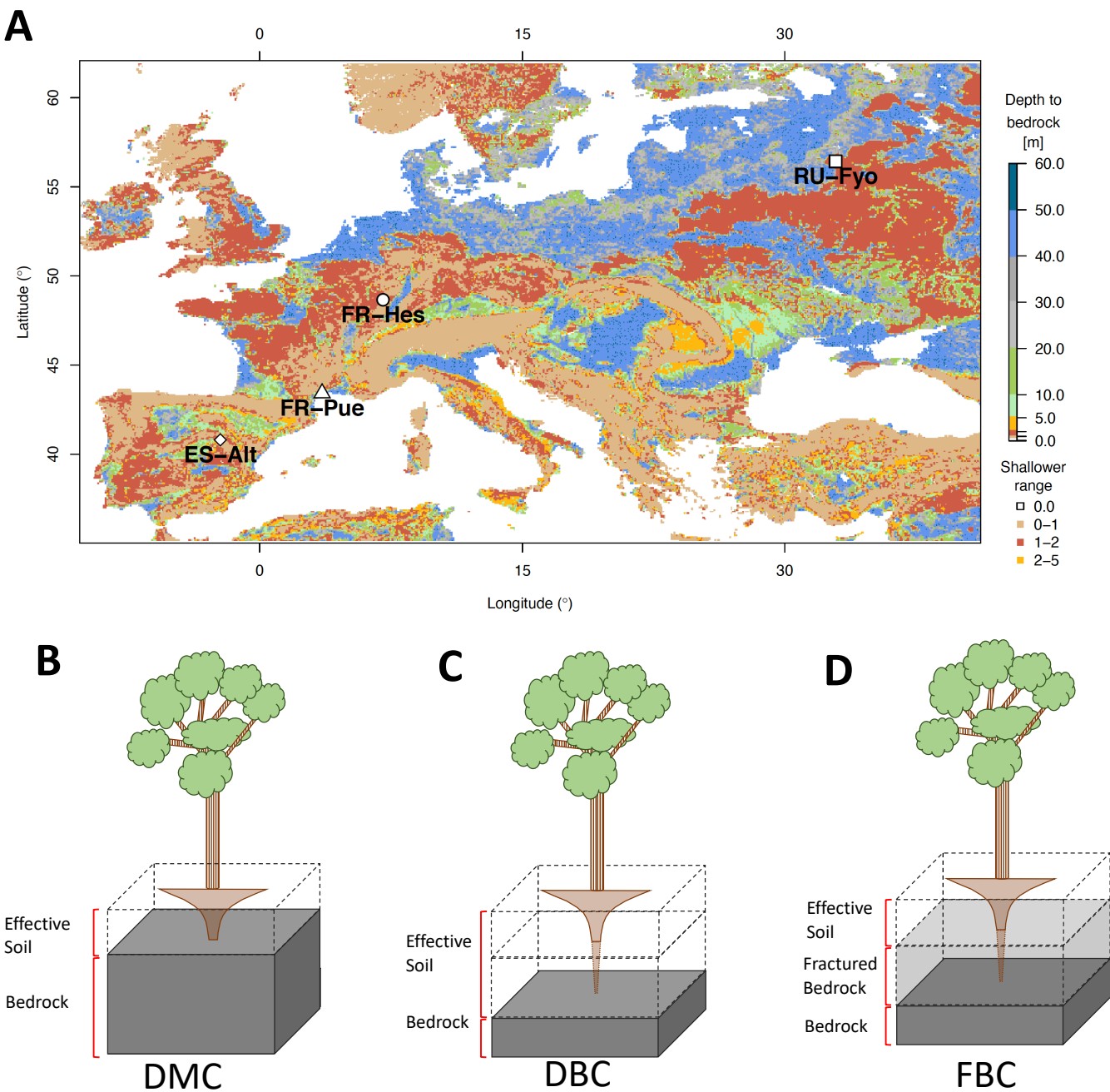

**Figure 1.** Geographical location of the experimental sites and spatial distribution of the depth to bedrock across Europe (A) based on Pelletier et al. (2016). The graphics below the map are the schematic of the three model configurations used in this work: default model configuration, DMC (B); deeper bedrock configuration, DBC (C); and fractured bedrock configuration, FBC (D). The block with dashed lines represents the soil profile, the solid grey block represents the impermeable bedrock layer as it is assumed by CLM 5, and the translucid grey block represents the mimicked fractured bedrock layer.

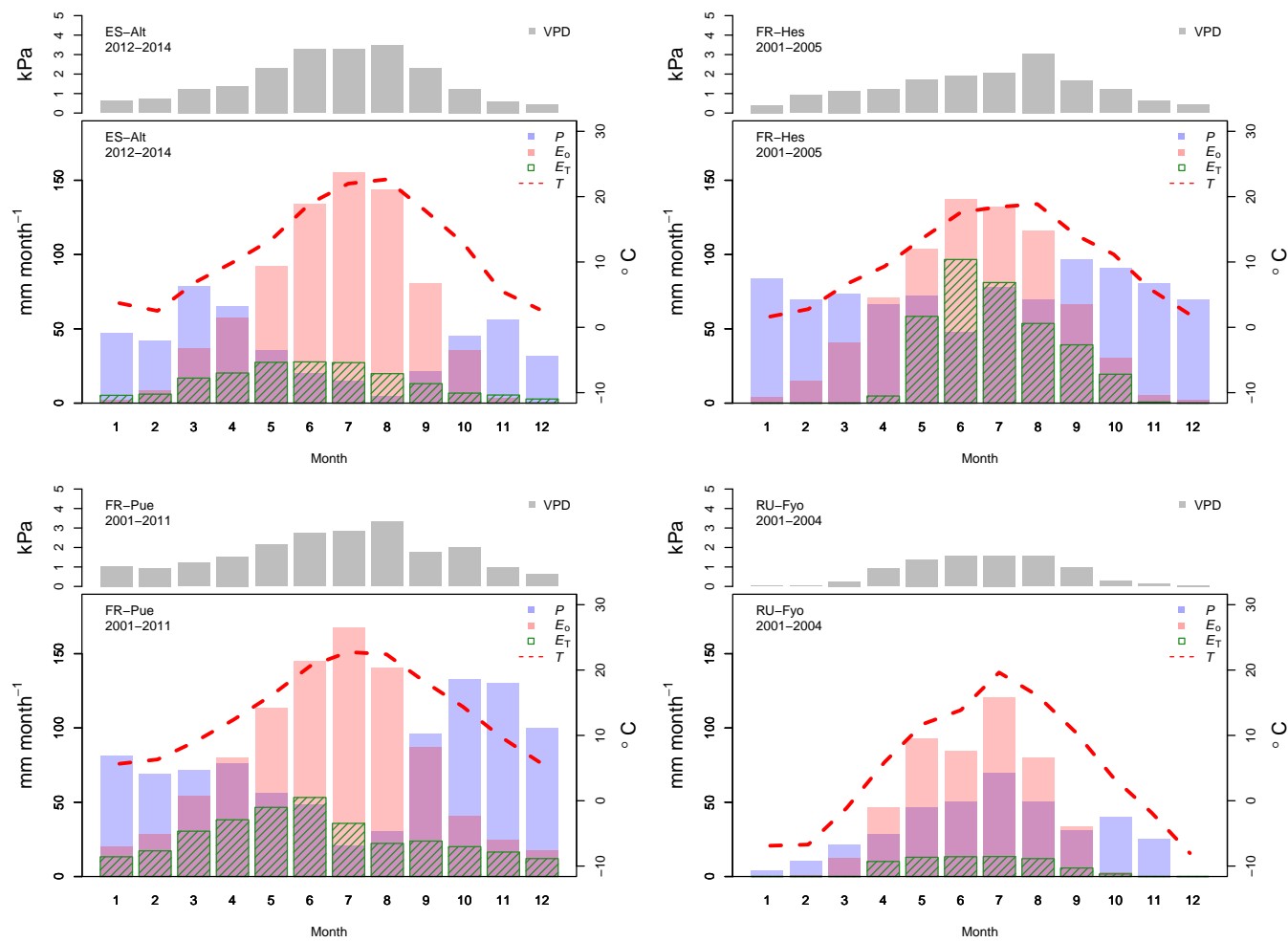

**Figure 2.** Temporal variation of the maximum vapor pressure deficit (VPD), total precipitation ($P$), total potential evaporation ($E_o$), total transpiration ($E_T$), and mean air temperature ($T$) for the selected experimental sites across Europe. The monthly values are based on the different sampling periods for each site.

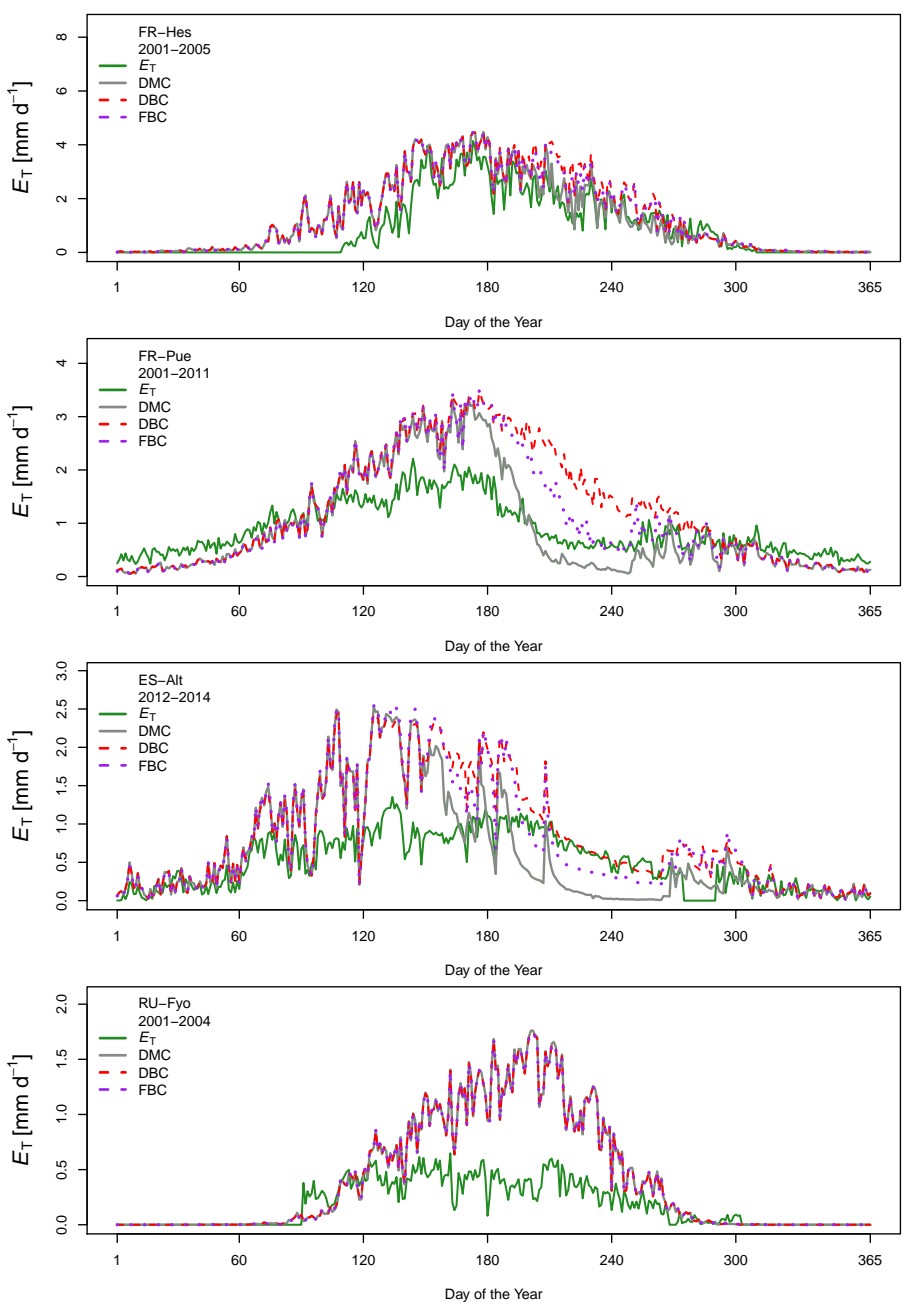

**Figure 3.** Median of the measured and modeled daily transpiration rates for the different model configurations at each experimental site.

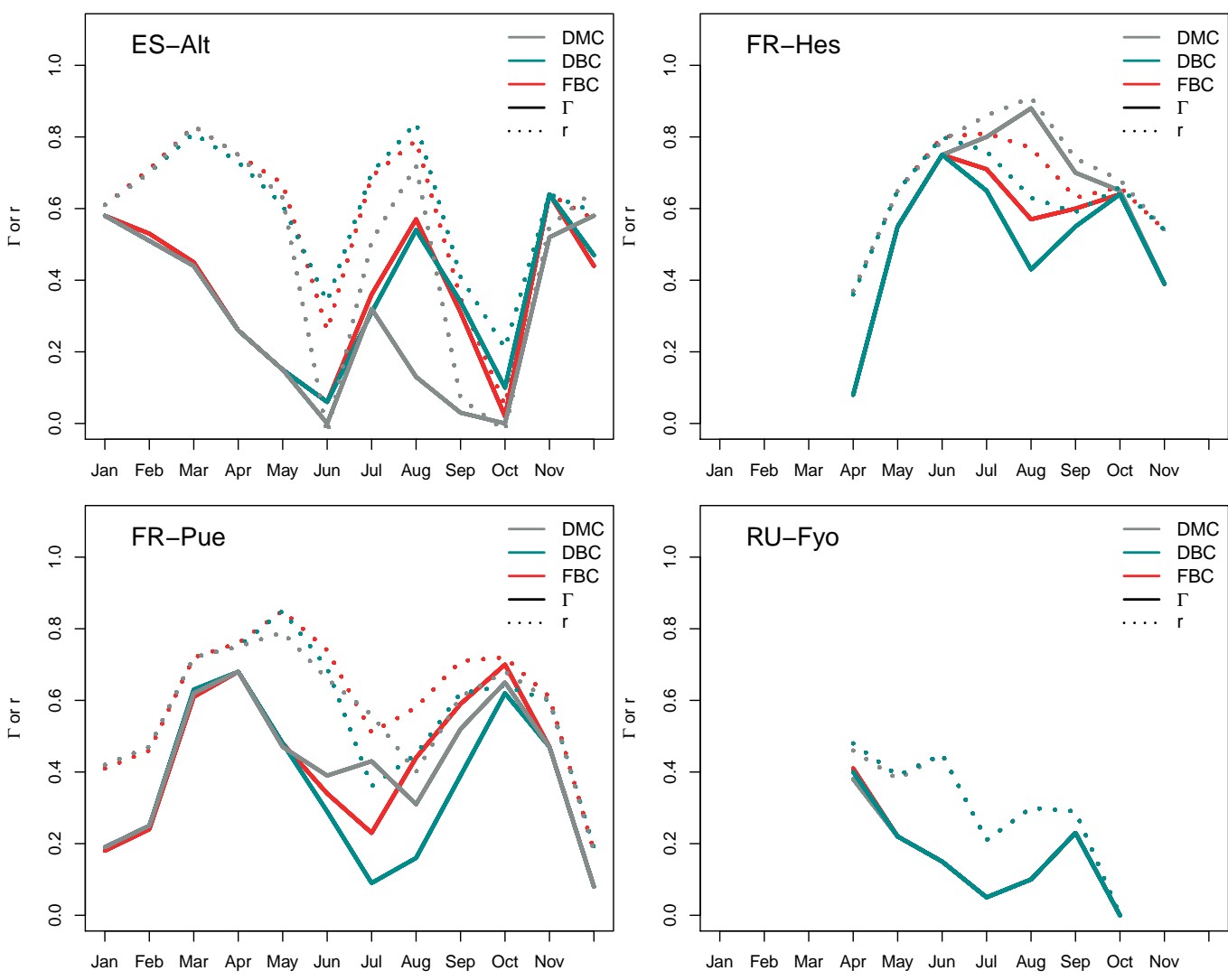

**Figure 4.** Multi-annual monthly variation of the Pearson correlation coefficient ($r$) and the index of agreement ($\Gamma$) for the default model configuration (DMC), deeper bedrock configuration (DBC), and fractured bedrock configuration (FBC). See Table A2 for the detailed list of $r$, $\alpha$, and $\Gamma$ values.

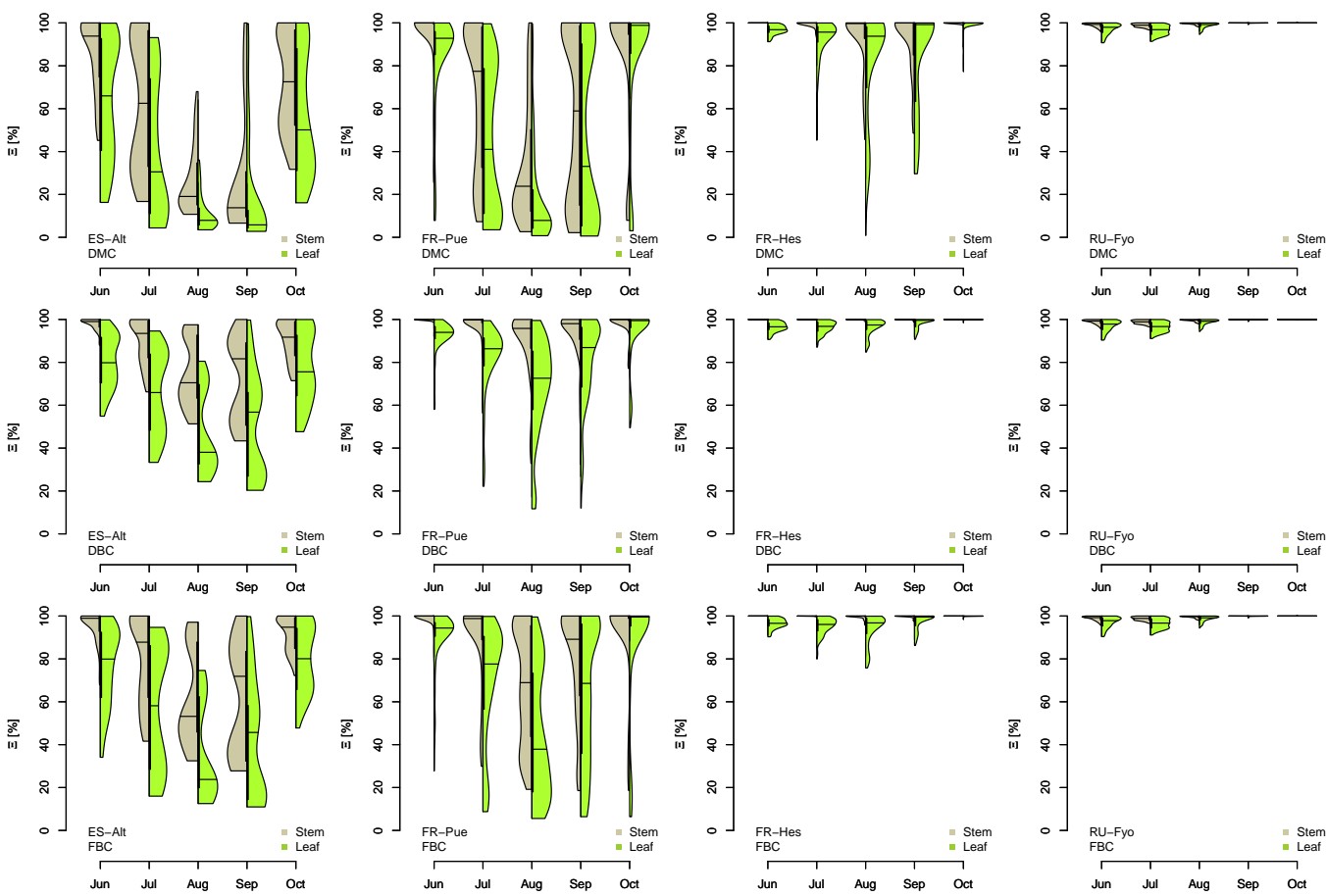

**Figure 5.** Hydraulic stress experienced by the modeled vegetation per experimental site based on the vulnerability curves of each plant organ (See Fig. A1 to Fig. A6) between June and October. Each vioplot describes the distribution of the hydraulic stress experienced by stem xylem (X) and sunny leaves xylem (L) expressed as percentage of conductivity ($\Xi$). Each vioplot is a visual representation of the probability density of the data, with the width and length representing the frequency and data distribution, respectively. The horizontal line corresponds to the median value and the thicker vertical line represents the range between first and third quartiles of the data.

**Table A2.** Monthly summary of the Pearson correlation coefficient (r), alpha coefficient ($\alpha$), and symmetry index ($\Gamma$) for the three model configurations and four experimental sites.

| Model Configuration | Month | ES-Alt | | | FR-Pue | | | FR-Hes | | | RU-Fyo | | |
|---|---|---|---|---|---|---|---|---|---|---|---|---|---|
| | | r | $\alpha$ | $\Gamma$ | r | $\alpha$ | $\Gamma$ | r | $\alpha$ | $\Gamma$ | r | $\alpha$ | $\Gamma$ |
| DMC | 1 | 0.60 | 0.95 | 0.57 | 0.43 | 0.45 | 0.19 | 0.00 | 0.00 | 0.00 | 0.00 | 0.00 | 0.00 |
| | 2 | 0.73 | 0.71 | 0.52 | 0.47 | 0.52 | 0.25 | 0.00 | 0.00 | 0.00 | 0.00 | 0.00 | 0.00 |
| | 3 | 0.81 | 0.52 | 0.42 | 0.72 | 0.86 | 0.62 | 0.00 | 0.00 | 0.00 | 0.32 | 0.63 | 0.20 |
| | 4 | 0.75 | 0.36 | 0.27 | 0.76 | 0.90 | 0.69 | 0.44 | 0.23 | 0.10 | 0.71 | 0.66 | 0.47 |
| | 5 | 0.62 | 0.24 | 0.15 | 0.79 | 0.60 | 0.47 | 0.69 | 0.85 | 0.58 | 0.44 | 0.70 | 0.31 |
| | 6 | -0.07 | 0.00 | 0.00 | 0.66 | 0.59 | 0.39 | 0.79 | 0.95 | 0.75 | 0.65 | 0.41 | 0.27 |
| | 7 | 0.47 | 0.61 | 0.29 | 0.57 | 0.77 | 0.44 | 0.86 | 0.91 | 0.79 | 0.19 | 0.27 | 0.05 |
| | 8 | 0.68 | 0.17 | 0.12 | 0.41 | 0.75 | 0.31 | 0.90 | 0.96 | 0.86 | 0.37 | 0.45 | 0.16 |
| | 9 | 0.07 | 0.45 | 0.03 | 0.61 | 0.86 | 0.53 | 0.74 | 0.95 | 0.70 | 0.35 | 0.95 | 0.34 |
| | 10 | -0.03 | 0.00 | 0.00 | 0.68 | 0.96 | 0.65 | 0.68 | 0.97 | 0.66 | 0.23 | 0.53 | 0.12 |
| | 11 | 0.55 | 0.94 | 0.52 | 0.61 | 0.77 | 0.47 | 0.55 | 0.62 | 0.34 | 0.00 | 0.00 | 0.00 |
| | 12 | 0.62 | 0.88 | 0.55 | 0.18 | 0.44 | 0.08 | 0.00 | 0.00 | 0.00 | 0.00 | 0.00 | 0.00 |
| DBC | 1 | 0.61 | 0.95 | 0.57 | 0.42 | 0.46 | 0.19 | 0.00 | 0.00 | 0.00 | 0.00 | 0.00 | 0.00 |
| | 2 | 0.72 | 0.71 | 0.52 | 0.47 | 0.53 | 0.25 | 0.00 | 0.00 | 0.00 | 0.00 | 0.00 | 0.00 |
| | 3 | 0.79 | 0.53 | 0.42 | 0.72 | 0.86 | 0.63 | 0.00 | 0.00 | 0.00 | 0.32 | 0.64 | 0.20 |
| | 4 | 0.74 | 0.36 | 0.27 | 0.76 | 0.90 | 0.69 | 0.44 | 0.23 | 0.10 | 0.72 | 0.69 | 0.49 |
| | 5 | 0.60 | 0.24 | 0.14 | 0.85 | 0.57 | 0.48 | 0.69 | 0.85 | 0.58 | 0.45 | 0.70 | 0.32 |
| | 6 | 0.29 | 0.16 | 0.05 | 0.69 | 0.41 | 0.28 | 0.79 | 0.95 | 0.76 | 0.65 | 0.41 | 0.27 |
| | 7 | 0.66 | 0.42 | 0.28 | 0.38 | 0.25 | 0.10 | 0.77 | 0.84 | 0.64 | 0.19 | 0.27 | 0.05 |
| | 8 | 0.81 | 0.62 | 0.50 | 0.46 | 0.36 | 0.17 | 0.63 | 0.65 | 0.41 | 0.37 | 0.45 | 0.17 |
| | 9 | 0.42 | 0.86 | 0.36 | 0.62 | 0.63 | 0.39 | 0.60 | 0.92 | 0.55 | 0.36 | 0.95 | 0.34 |
| | 10 | 0.22 | 0.48 | 0.10 | 0.63 | 0.97 | 0.62 | 0.66 | 0.98 | 0.65 | 0.24 | 0.53 | 0.12 |
| | 11 | 0.66 | 0.99 | 0.65 | 0.61 | 0.78 | 0.47 | 0.55 | 0.62 | 0.34 | 0.00 | 0.00 | 0.00 |
| | 12 | 0.56 | 0.78 | 0.44 | 0.19 | 0.45 | 0.08 | 0.00 | 0.00 | 0.00 | 0.00 | 0.00 | 0.00 |
| FBC | 1 | 0.60 | 0.96 | 0.58 | 0.42 | 0.44 | 0.19 | 0.00 | 0.00 | 0.00 | 0.00 | 0.00 | 0.00 |
| | 2 | 0.73 | 0.73 | 0.53 | 0.46 | 0.51 | 0.24 | 0.00 | 0.00 | 0.00 | 0.00 | 0.00 | 0.00 |
| | 3 | 0.82 | 0.52 | 0.43 | 0.72 | 0.85 | 0.61 | 0.00 | 0.00 | 0.00 | 0.31 | 0.63 | 0.20 |
| | 4 | 0.75 | 0.35 | 0.27 | 0.77 | 0.90 | 0.69 | 0.44 | 0.23 | 0.10 | 0.71 | 0.70 | 0.49 |
| | 5 | 0.66 | 0.23 | 0.15 | 0.85 | 0.56 | 0.48 | 0.69 | 0.85 | 0.58 | 0.45 | 0.70 | 0.32 |
| | 6 | 0.21 | 0.22 | 0.05 | 0.74 | 0.46 | 0.34 | 0.80 | 0.96 | 0.76 | 0.65 | 0.41 | 0.27 |
| | 7 | 0.65 | 0.51 | 0.33 | 0.52 | 0.44 | 0.23 | 0.81 | 0.86 | 0.69 | 0.19 | 0.27 | 0.05 |
| | 8 | 0.76 | 0.69 | 0.53 | 0.58 | 0.76 | 0.44 | 0.77 | 0.72 | 0.55 | 0.37 | 0.45 | 0.17 |
| | 9 | 0.35 | 0.92 | 0.33 | 0.71 | 0.84 | 0.60 | 0.64 | 0.94 | 0.60 | 0.36 | 0.95 | 0.34 |
| | 10 | 0.05 | 0.46 | 0.03 | 0.72 | 0.97 | 0.70 | 0.66 | 0.98 | 0.65 | 0.24 | 0.53 | 0.12 |
| | 11 | 0.65 | 1.00 | 0.65 | 0.61 | 0.77 | 0.47 | 0.55 | 0.62 | 0.34 | 0.00 | 0.00 | 0.00 |
| | 12 | 0.54 | 0.76 | 0.41 | 0.18 | 0.42 | 0.08 | 0.00 | 0.00 | 0.00 | 0.00 | 0.00 | 0.00 |

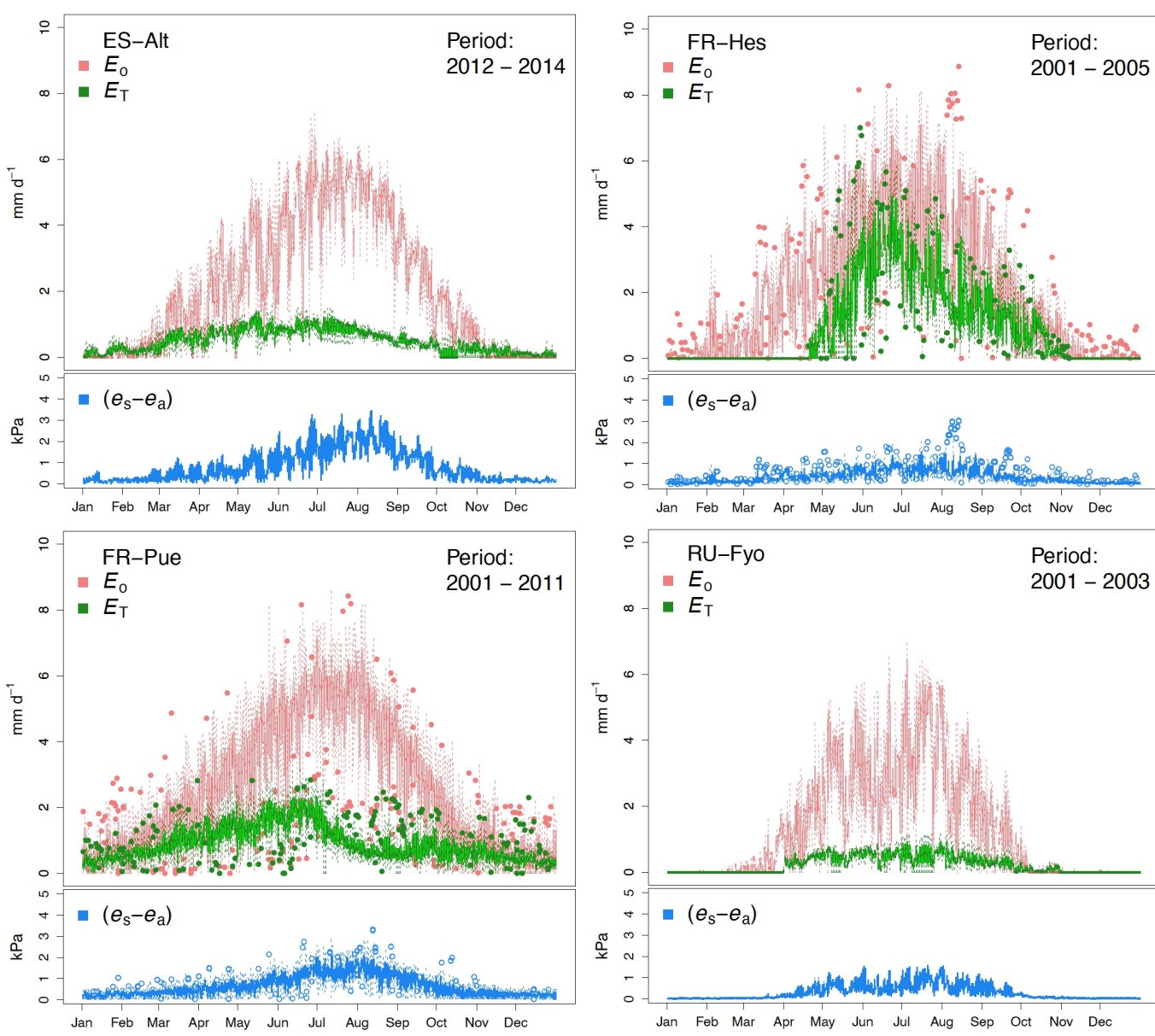

**Figure A1.** Multi-annual daily boxplots for potential evaporation ($E_o$), stand transpiration ($E_T$), and vapor pressure deficit ($\Lambda$) of the selected experimental sites across Europe. The boxplot represents the data contained between the first and third quartiles, the central line is the median, the whiskers represent a predefined distance from the median (1.5 x inter-quartile range), and the dots are the outliers.

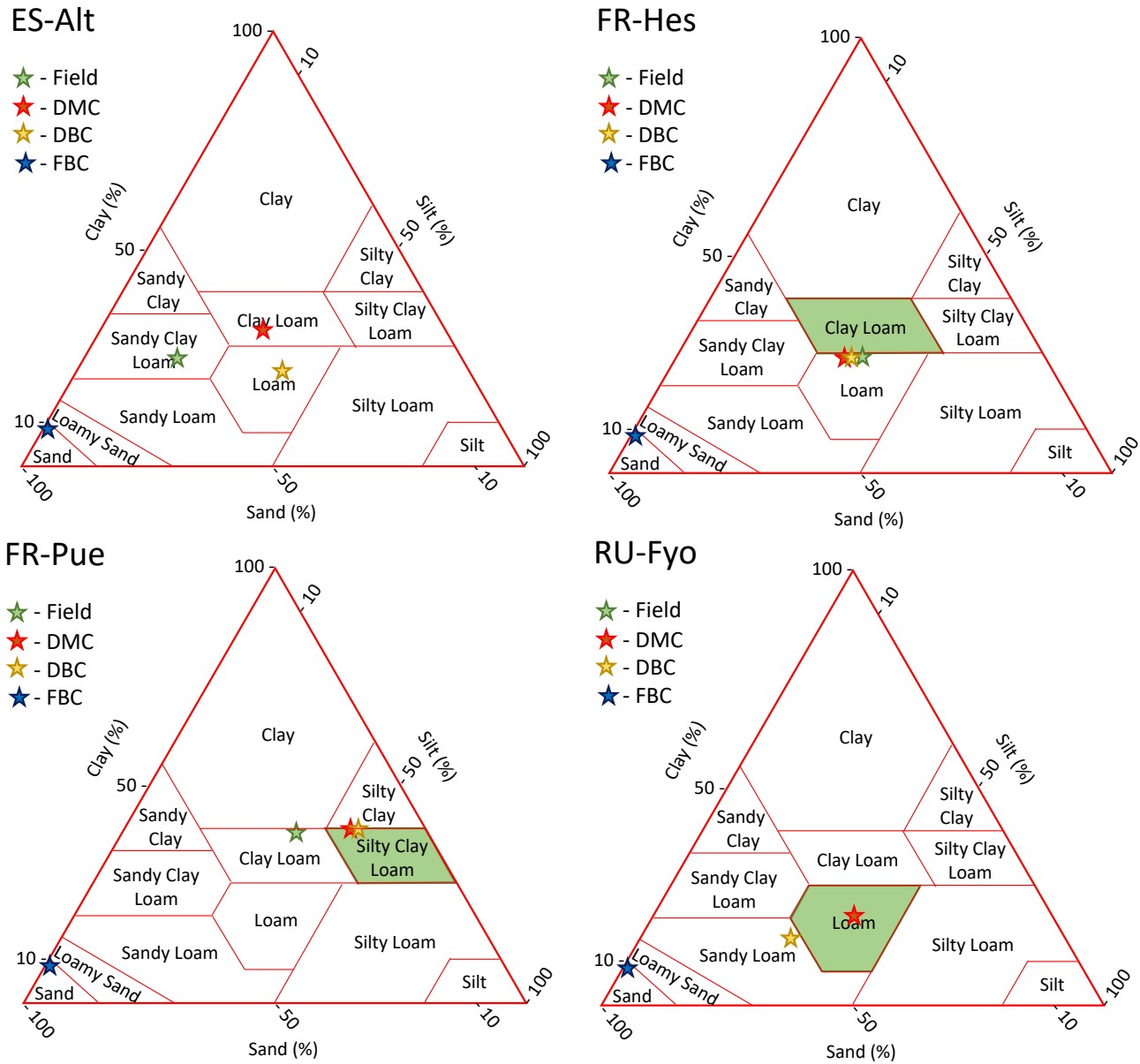

**Figure A2.** Soil texture classification of each experimental site and model configuration. The green star and polygons highlight the experimental site classification according to the literature (the classification may differ between authors). Red, yellow, and blue stars represent the substrata classification according to the default, deeper bedrock, and fracture bedrock configurations, respectively.

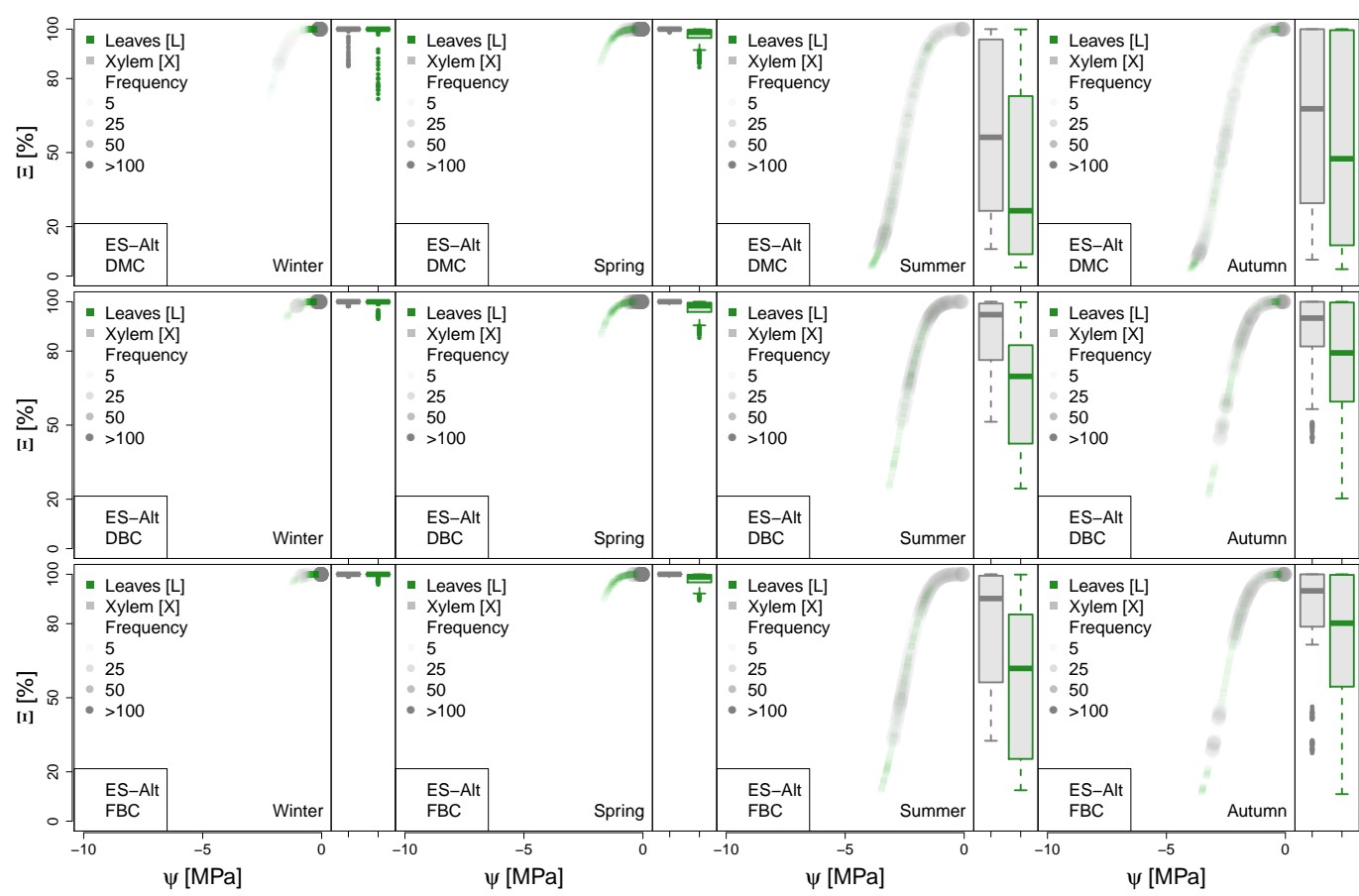

**Figure A3.** Seasonal variation of the plant vulnerability curves describing the hydraulic stress as percentage of conductivity (Ξ) experienced by the modeled vegetation in ES-Alt experimental site. Each plot describes the hydraulic stress experienced by stem xylem (X) and sunny leaves xylem (L).

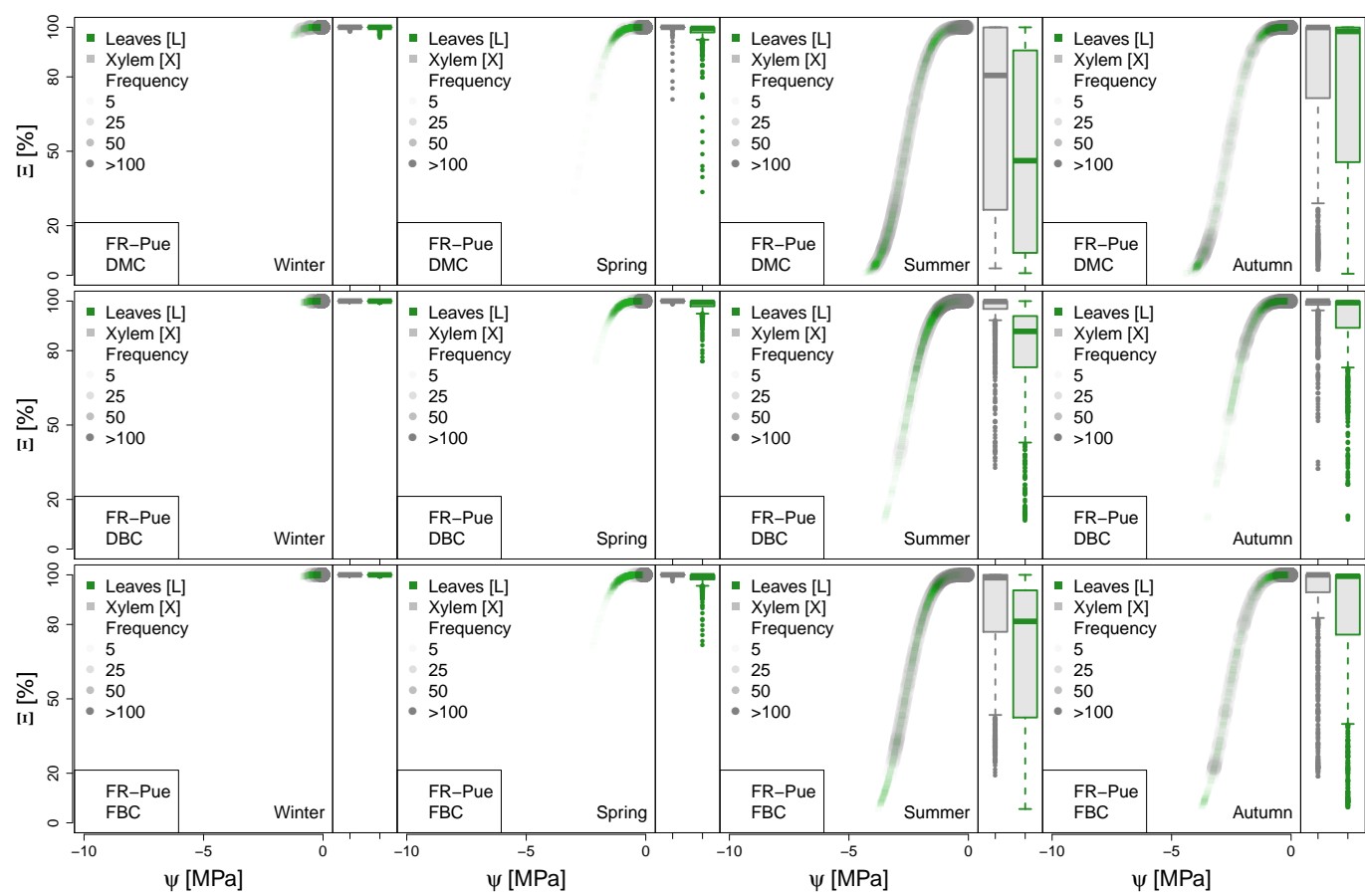

**Figure A4.** Seasonal variation of the plant vulnerability curves describing the hydraulic stress as percentage of conductivity (Ξ) experienced by the modeled vegetation in FR-Pue experimental site. Each plot describes the hydraulic stress experienced by stem xylem (X) and sunny leaves xylem (L).

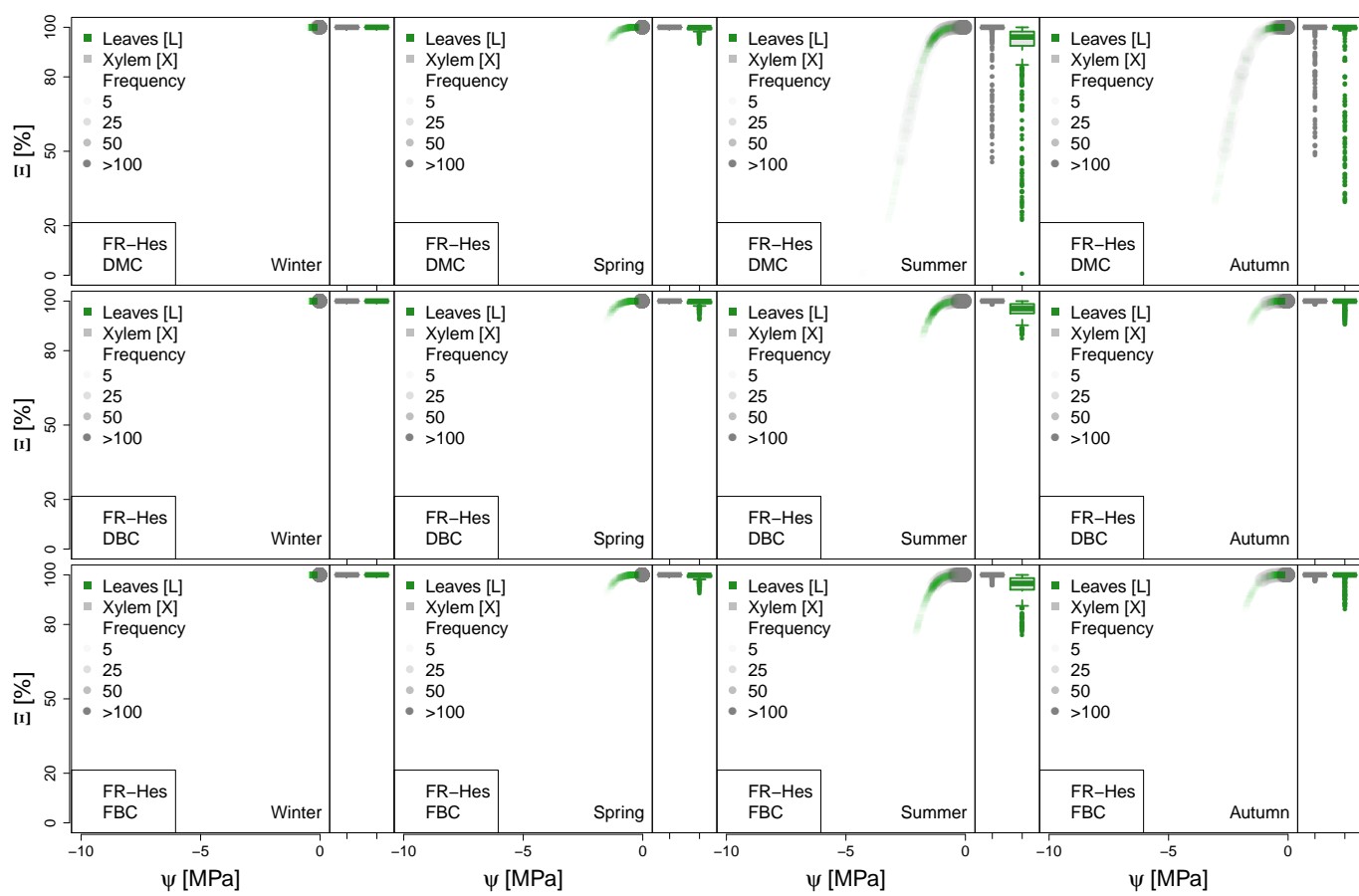

**Figure A5.** Seasonal variation of the plant vulnerability curves describing the hydraulic stress as percentage of conductivity (Ξ) experienced by the modeled vegetation in FR-Hes experimental site. Each plot describes the hydraulic stress experienced by stem xylem (X) and sunny leaves xylem (L).

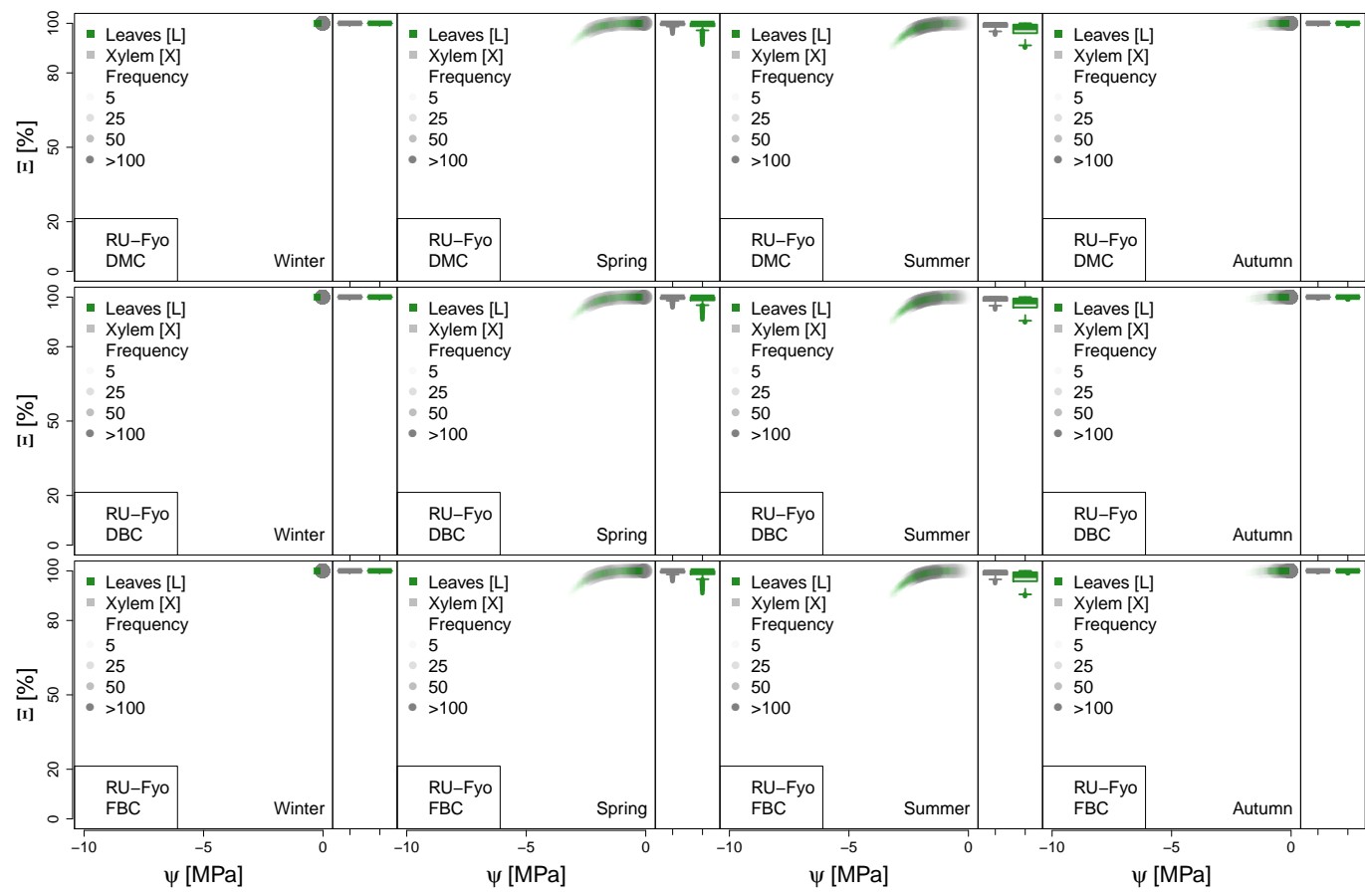

**Figure A6.** Seasonal variation of the plant vulnerability curves describing the hydraulic stress as percentage of conductivity (Ξ) experienced by the modeled vegetation in RU-Fyo experimental site. Each plot describes the hydraulic stress experienced by stem xylem (X) and sunny leaves xylem (L).