# Peer review of "Exploring the role of bedrock representation on plant transpiration response during dry periods at four forested sites in Europe"

_Biogeosciences, 2021_

## Author Comment (AC1)

**Reply to comments of Reviewer 1**

In *"blue"* we copied the comments of the reviewer, in **black** our reply, in **green** new additions that will be included in the manuscript, and in **red** the deletions to the manuscript. We also indicated within square brackets the line number when we refer to specific points or sections of the manuscript "[Line: ]". The numbering of figures and tables of this reply are preceded by R aiming to differentiate them from the original figures and tables of the manuscript.

*"Paper summary: In response to a growing body of research indicating that plants routinely use water from bedrock, these authors asked the question: what happens to modeled plant transpiration if, instead of relying exclusively on soil water, they are allowed to access a deeper bedrock bucket? They found that having access to more water improved the accuracy of transpiration in a widely used land surface model (when compared to actual sap flow data) in places with pronounced dry seasons. The authors suggest that this provides additional motivation for the better inclusion of plant-available bedrock water in land surface models. The manuscript is well written and easy to read."*

**1. Reply:**

We would like to thank the reviewer's positive comments and the constructive feedback. Hereafter, we provide a separate response to each query.

*"I am supportive of the goals of this manuscript and would like to see it published, but I would also appreciate the authors considering how they could address what I perceive are two shortcomings:*

> 1. *The study illustrates transpiration dynamics using field data for sapflow at four sites, but no actual local field information about the storage dynamics of the bedrock underlying soil, local rooting profiles, etc. is provided. So, there is little meaningful context regarding the subsurface properties at the sites (properties that are the primary focus of the paper)."*

**2. Reply:**

Detailed physical features such as hydraulic conductivity, water holding capacity, or percentage of fractures are not available for the weathered bedrock beneath the experimental sites. There have been some regional assessments of the role played by the interaction between soil and groundwater systems close to Puechabon [FR-Pue] and Hesse [FR-Hes] (Kirchen et al. 2017, Ollivier et al. 2021) but detailed information of their physical properties is missing. However, each location has different degrees of data availability describing soil characteristics. The sites Puechabon [FR-Pue] (e.g., Hoff et al. . 2002), Hesse [FR-Hes] (e.g., Granier et al. . 2008), and Alto Tajo [ES-Alt] (e.g., Forner et al. 2018) have a good record, but the information available for Fyorodorovskoye [RU-Fyo] is limited (e.g., Vygodskaya et al. 2002). Table R1 summarizes the available information and

provides a better context of the subsurface structure of each experimental site. Therefore, aiming to provide a better contextualization of the subsurface characteristics of each site, we proposed to add Table R1 and improve the characterization of each site as follows:

In [Line: 95]:

"… and lowlands (RU-Fyo). All the sites lack detailed information concerning the physical properties of the weathered bedrock, such as hydraulic conductivity, water holding capacity, or percentage of fractures. The sites are distributed across an environmental …"

For ES-Alt [Line: 101]:

"… et al. 2017). The soils are formed from Cretaceous carbonate rocks settled on top of sandy sediments (Carcavilla et al. 2008), showing a poor development (Granda et al. 2012) with maximum soil thickness varying between 25 cm at the top of the slopes, and more than 1.0 m at the bottom (Martín-Moreno et al. 2014). The climate …"

[Line: 108]:

"… et al. 2014). Quercus ilex trees at this site have the capacity to allocate structural and fine roots down to 8.0 m depth (Penuelas et al. 2003). The plant …"

For FR-Pue [Line: 112]:

"… et al. 2002). The superficial soil layers have a high soil permeability thanks to the elevated stone fraction of 0.75. On the other hand, soil layers beneath 50 cm have a larger clay content (> 30%) and a stone fraction of more than 0.9 (Limousin et al. 2009, Pita et al. 2013). This location …"

[Line: 115]:

"… 2008). This forest stand allocates most of the fine root biomass in the top 50 cm of the soil profile with a small fraction of roots reaching depths down to 4.5 m (Allard et al. 2008). Similarly, to …"

For FR-Hes [Line: 112]:

"… 2007). It is a clear transition between the eluviated horizon and the horizon with high clay accumulation at 50 cm depth (Zapater et al. 2012), leading to a low clay content in the upper soil layers (Granier et al. 2000 a). It has … "

[Line: 124]:

"… 2001). Fagus sylvatica trees allocate most of the fine roots in the upper 40 cm of soil with some fine roots reaching depths down to 1.5 m (Granier et al. 2000a, Granier et al. 2000b, Zapater et al. 2012, Bestch et al. 2011). The dominant …"

For RU-Fyo [Line: 131]:

"… and still present. The peat layer at this site has an average depth of 50 cm (Schulze et al. 2002, Vygoskaya et al. 2002, Arneth et al. 2002, Kurbatova et al. 2002), and the glacial deposits result in a loamy texture of the soil beneath it (Novenko et al. 2010, Schulze et al. 2002). The water table at this site is shallow, forcing the trees to allocate most of the fine roots in the top 20 cm of the soil (Milyukova et al. Schulze et al. 2002, Vygodskaya et al. 2002). The climate …"

Also, adding Table R1 as an appendix:

Table R1. Summary of the main sub-surface characteristics of each experimental site.

| Main Site Characteristics | ES-Alt | FR-Pue | FR-Hes | RU-Fyo |
|---|---|---|---|---|
| Rooting Depth (m) | 8.0 | 4.5 | > 1.5 | 0.2 |
| Peat Layer Depth (cm) | None | None | None | 50 |
| Soil Depth (cm) | 20-40 \| 100 | 50 | 145 | N.A. |
| Soil Texture: % Clay | 25.9 | 39 | 25 | N.A. |
| Soil Texture: % Sand | 57.3 | 26 | N.A. | N.A. |
| Soil Texture: % Silt | 16.8 | 35 | N.A. | N.A. |
| Soil Type | Sandy Clay Loam | Silty Clay Loam | Clay Loam | Loam |
| Soil Permeability | N.A. | High | N.A. | N.A. |
| Superficial Stone Fraction | N.A. | 0.75 (0-50cm) | N.A. | N.A. |
| Deep Stone Fraction | N.A. | 0.90 (>50 cm) | N.A. | N.A. |
| Bedrock Type | Cretaceous carbonate | Jurassic Limestone | Sandstone | Glacial deposits |
| Water Table Depth | Deep | Deep | Deep | Superficial |
| Plant Water | Soil Water, Weathered Bedrock, Groundwater | Soil Water, Weathered Bedrock, Groundwater | Soil Water | Non-Saturated Substratum |
| Hydraulic Lift | Present | N.A. | Present | N.A. |
| References | Grossiord et al. (2015) Penuelas et al. (2003) Forner et al. (2018) Martin-Duque et al. (2008) | Allard et al. (2008) Pita et al. (2013) Limousin et al. (2009) | Granier et al. (2007) Granier et al. (2000a) Granier et al. (2000b) Le Goff et al. (2001) Zapater et al. (2012) Betsch et al. (2011) | Schulze et al. (2002) Vigodskaya et al. (2002) Novenko et al. (2010) Arneth et al. (2002) Kurbatova et al. (2002) Milyukova et al. (2002) |

*"This means the study essentially looked at the effect of varying a model parameter (water storage bucket size) on T and found that the default model configuration could be improved upon. Other default model parameters could have also been varied (the PFT properties, for example), and modeled transpiration might have been improved as well."*

**3. Reply:**

We agree with the reviewer's point that many environmental and physiological parameters of the model can be changed (e.g., $\Psi_{p50}$, $K_{max}$, soil texture, slope of the stomata conductance model, etc.) to improve the simulated transpiration response. However, the focus of this manuscript is not to tune the CLM model to each experimental site. Instead, we wanted to test the sensitivity of the simulated transpiration to the bedrock representation using novel sap flow data as a reference dataset. Consequently, we only focus on modifying the depth to bedrock using the default model to isolate its effect.

Moving the bedrock deeper implies solving the issue of how to parameterize this newly created water storage. "Section 3.2 Bedrock Configuration" [Line: 154] describes the selected approach which relies on two main assumptions: a fractured bedrock will have a high-water conductivity and low water holding capacity. This approach is based on the recent studies at regional and continental scales stressing the importance of the accumulated sediments in weathered bedrock fractures that enhances the water holding capacity beneath the soil (Jian et al. 2020), allowing the woody ecosystems to use the rock moisture at regional scales (McCormick et al. 2021). The widespread use of rock moisture by vegetation is supported by the vast number of studies carried out during the last two decades stressing the importance of adding the weathered bedrock as a small reservoir that serves as a buffer during dry conditions (Hubbert et al. 2001, Aranda et al. 2007, Baldocchi et al. 2010, David et al. 2013, Forner et al. 2018, Gil-Pelegrín et al. 2017, Nadezhdina et al. 2008, Penuelas & Filella 2003, Zapater et al. 2012).

Furthermore, numerous studies have shown that the increasing complexity of LSMs may also increase the number of model parameters with substantially uncertain values (e.g., Mu et al. 2021). This trend requires the implementation of statistically robust approaches to test model sensitivity and uncertainty. A prominent example along this line is the Parameter Perturbation Ensemble Project using CLM5 (Lawrence et al. 2020). We recognize that all these points should be clearer in the manuscript. Consequently, we propose to add:

In [Line 49]:

> "… and Koven, 2020). However, the recent developments in LSMs have increased the number of parameters exerting control over multiple fluxes, pointing out the need to provide a better way to constrain these parameters to improve the LSM simulations (Mu et al. 2021). These constraints are of high importance because they allow assessing the effect of the parameter selection on the model sensitivity and uncertainty (Lawrence et al. 2020). The representation of subsurface …"

In [Line: 76]:

> "… et al. 2016). Recent studies stressed the importance of the accumulated sediments in weathered bedrock fractures that enhances the water holding capacity beneath the soil (Jian et al. 2020). These sediments allow the woody ecosystems to use the rock moisture at the continental scale across different rock

types (McCormick et al. 2021). The widespread use of rock moisture by vegetation is supported by the vast number of studies carried out during the last two decades stressing the importance of adding the weathered bedrock as an additional water reservoir that serves as a buffer water supply during dry conditions (Hubbert et al. 2001, Aranda et al. 2007, Baldocchi et al. 2010, David et al. 2013, Forner et al. 2018, Gil-Pelegrín et al. 2017, Nadezhdina et al. 2008, Penuelas & Filella 2003, Zapater et al. 2011). This reservoir is critical for properly representing transpiration fluxes during dry periods (e.g., summer, drought, heat waves) of vegetation growing in water-limited environments. Consequently, it is necessary to provide a preliminary evaluation of the impact that additional water storage may have on the simulated transpiration fluxes."

Also, aiming to clarify the decision of using the default configuration, we proposed to add the following:

In [Line 145]:

"… et al.  2019). We use the default plant physiological parameterization (Lawrence et al.  2019) according to the PFT classification of each experimental site. Site-specific …"

In [Line 166]:

"… clay (Table 1). This approach aims to mimic the hydrological behavior of a fractured bedrock based on two main assumptions: a fractured bedrock should have a high-water conductivity and low water holding capacity. The high sand percentage will mimic the fast water movement through the primary and secondary porosity of the fractured bedrock. At the same time, the low clay content allows having a low water holding capacity for plant water uptake."

Also, we will add the reference "(Jones & Graham 1993)" in Line 27 to support the sentence.

The soil parametrization used for the Default Model Configuration (DMC) and Deeper Bedrock Configuration (DBC) agreed with the published data for FR-Hes, FR-Pue, and FR-Hes (Figure R1). These configurations are located within the boundaries of the soil texture-classification of each site. In this regard, the agreement of soil water storage capacity, infiltration, and percolation rates will agree with the expected site conditions. Clay content in ES-Alt is similar among model configurations, providing a similar water holding capacity. However, the sand content in the model for the DMC and DBC configurations determines a reduced hydraulic conductivity than the one expected from the field data. This condition implies that the CLM 5 model tends to percolate less water than under actual conditions. In this regard, the subsurface parametrization doesn't need to be adjusted for the sites, while the infiltration and percolation rates of ES-Alt are expected to be larger than the modeled results.

Aiming to clarify the impacts of soil characteristics on the soil water reservoir, we proposed to add Figure R1 as an appendix and the following sentences:

In [Line: 265]:
    "… configurations.

    The soil parametrization used in DMC and DBC agreed with the published data for FR-Hes, FR-Pue, and FR-Hes (Figure R1). These configurations are located within the boundaries of the soil texture classification of each site. In this regard, the agreement of soil water storage capacity, infiltration, and percolation rates agreed with the expected site conditions. Clay content in ES-Alt is similar between model configurations and site conditions, providing a similar water holding capacity. However, the sand content of the model for DMC and DBC configurations represents a lower hydraulic conductivity than the one expected for this site."

[Figure]

Figure R1. Soil texture classification of each experimental site and model configuration. The green star and polygons highlight the experimental site classification

according to the literature (the classification may differ between authors). Red, yellow, and blue stars represent the substrata classification according to the default, deeper bedrock, and fracture bedrock configurations, respectively.

*"So, while the authors have shown that changing a model parameter from the default can improve model performance (larger storage buckets can improve T representation [and I don't doubt that this is the likely reason]), without any actual data showing that plants use deeper water from bedrock at these sites it has not been demonstrated that this is mechanistically why T has improved for these particular sites. Is any of this context available at the four study sites, and could it be added to the paper?"*

**4. Reply:**

The reviewer highlights the lack of evidence supporting the deep-water use by trees at the selected sites, especially at Puechabon [FR-Pue] and Alto Tajo [ES-Alt]. We agree with the reviewers that such evidence would be very valuable, but unfortunately, it is not available for all sites. However, there is extensive evidence at the species level, supporting the assumption that different oak species can access deep water storage (Aranda et al. 2007, Baldocchi et al. 2010, David et al. 2013, Forner et al. 2018, Gil-Pelegrín et al. 2017, Nadezhdina et al. 2008, Penuelas & Filella 2003, Zapater et al. 2011). Baldocchi et al. (2010) and Forner et al. (2018) show evidence of deep-water uptake by Quercus ilex trees growing at Puechabon and Alto Tajo, respectively.

Fagus sylvatica trees are believed to strongly rely on shallow soil water because of their superficial root system (Houston Durrant et al. 2016, Kirchen et al. 2017, Leuschner 2020), using in some cases more than 40 % of water allocated in the first 8 cm of soil (Lüttschwager & Jochheim 2020). Fagus sylvatica trees growing in Hesse (France) have shown a strong reliance on shallow soil water (Granier et al. 2000b, Zapater et al. 2012) because of the high accumulation of fine root biomass on the first 50 cm of soil (Le Goff & Ottorini 2001, Granier et al. 2008), and the stable water isotope evidence of no deep-water use (Zapater et al. 2012).

The rooting characteristics of Picea abies, Pinus sylvestris, and Betula pubescens (Gale & Grigal 1987, Špulák et al. 2021) might allow the trees at Fyodorovskoye to retrieve water from different depths. However, the transpiration rates at this site are reduced, likely because of the shallow water table and anoxic conditions in the root zone (Kurbatova et al. 2002, Launiainen et al. 2016). [Lines: 449-451].

Aiming to address this issue, we propose to add the following in the manuscript:

In [Line: 373]:

"... Dietricht, 2018). Oak tree species are known to access deep water storage because of their extensive rooting depths (Gil-Pelegrín et al. 2017). The Quercus ilex trees growing at ES-Alt and FR-Pue have shown this feature (Baldocchi et al. 2010, Forner et al. 2018) allowing the trees to transpire during the dry season despite the low soil water potentials. The transpiration signal ..."

In [Line: 378]:

> "... transpired soil water. Fagus sylvatica trees strongly rely on shallow soil water because of their superficial root system (Lüttschwager & Jochheim 2020), a condition documented at FR-Hes by Granier et al. (2000b) and Zapater et al. (2012). Meanwhile, during drought periods, this tree species uses the water stored in the trunk and roots as a reservoir to maintain transpiration until the next rainfall event (Betsch, et al. 2011). However, the limited access of vegetation to ..."

> *"Based on the findings of the paper, what should be done by the modeling community?"*

**5. Reply:**

Previous studies have highlighted the importance of groundwater and lateral flow for an improved simulation of transpiration fluxes in LSMs. In contrast, other studies highlighted the importance of an extended rooting system with the same aim. However, the binary distinction between soil and bedrock and the neglect of any water storage below the soil layer may prevent the LSM from producing more accurate results in locations with shallow soils during dry periods. In this work, we show that allowing the root system to access water stored in the weathered bedrock improves the transpiration and plant water stress estimates during dry periods. We think that the modeling community should address these two issues in a unified approach, which will eventually improve the representation of water supply at sites with shallow soils and pronounced dry periods.

Consequently, the discussion will be improved adding the following in [Line: 152]:

> "5.3 Breaking the bedrock to release the roots
>
> The occurrence and severity of extreme weather conditions like droughts and heat waves (He et al. 2020) highlight the importance of representing the vegetation's mechanisms to avoid or cope with the adverse effects of this new reality. However, the inclusion of rock moisture stored in the weathered bedrock, which is neither soil water nor groundwater, is necessary to simulate plant transpiration under limited water conditions (Rempe and Dietrich, 2018). The challenge to quantify this rock moisture is linked with the large uncertainty in determining the physical characteristics of this weathered bedrock (Pelletier et al. 2016) and the lack of spatially distributed field information related to the water storage properties of the weathered bedrock. Previous studies have highlighted the importance of groundwater and lateral flow for an improved simulation of transpiration fluxes in LSMs (Maxwell and Condon, 2016, Zeng et al. 2018). In contrast, other studies highlighted the importance of an extended rooting system with the same aim (Fan et al. 2017, Ichii et al. 2009). However, the focus of LSMs on soil water neglects the interaction between these essential components and the weathered bedrock. In this work, we show that allowing the root system to access water stored in the weathered bedrock improves the transpiration and plant water stress estimates during dry periods. The modeling community should address these two issues in a unified approach, eventually improving the water supply at sites with shallow soils

and dry conditions. This unified approach, where we allow to break the bedrock and release the root profile, will create a new water reservoir that will refine the vulnerability assessment of forest ecosystems growing in regions with a tendency to experience drier conditions.*"*

*"Should the water storage bucket just be freely calibrated instead of prescribed?"*

**6. Reply:**

We think that the free calibration process carries the general danger of getting "the right results for the wrong reasons". It could be a suitable solution for point-scale simulations if field data is available (e.g., electro resistivity measurements, well description logs) to provide physical constraints for the water holding capacity and vertical extent of the fractured bed rock. However, for an application of the land surface model at the regional or global scale, this bucket should be prescribed accordingly with the spatial distribution of the pedology, geology, and climate conditions driving the process of bedrock weathering.

*"What exactly is the goal of changing this parameter? To improve accuracy of historically observed T, or to better predict T under non-stationary climate, etc?"*

**7. Reply:**

Both. If the model is not capable of reproducing dry season transpiration in the past, it is likely not suitable to predict transpiration in future climate conditions. The main goal is to allow the model to better represent the plant water uptake of ecosystems relying on rock moisture by including this missing water storage (e.g., Mediterranean forests, dry lands). This is of high relevance for the non-stationary climate conditions expected under different climate change scenarios. Despite the scenario, the expected seasonal changes in intensity and frequency of precipitation and drought (Grillakis 2019, Hosseinzadehtalaei et al. 2020) will induce drier soil conditions. Consequently, it is of paramount importance to have a better representation of all the water storages that alleviate the soil water deficits. In this way, it will be possible to identify those areas/locations where the rock moisture will become more important for the established vegetation.

We propose to add the following in [Line: 33]:

"… et al. 2014). The inclusion of rock moisture stored in the weathered bedrock can better represent dry season vegetation water use at sites with shallow soils. Furthermore, this additional water reservoir may become even more significant under climate change, where seasonal changes in intensity and frequency of precipitation and drought (Hosseinzadehtalaei et al. 2020, Grillakis 2019) are expected to result in extended drought severity and duration. Consequently, it is paramount to better represent the entire water storage accessible to plants during dry periods. However, due to the …"

2. *"Other studies have already shown that increasing the size of the storage bucket accessible to plants can improve modeled T patterns in seasonally dry (e.g., Mediterranean) climates (e.g., Ichii, K., Wang, W., Hashimoto, H., Yang, F., Votava, P., Michaelis, A. R., & Nemani, R. R. [2009]. Refinement of rooting depths using satellite-based evapotranspiration seasonality for ecosystem modeling in California. Agricultural and Forest Meteorology, 149(11), 1907-1918.). Yes, these studies do this by changing rooting depths, or adding deeper soil (rather than calling it bedrock), but isn't the fundamental result the same: more stored water accessible to plants? What exactly is the novel finding in this study in relation to what these other studies have done (which is to change a model parameter that ultimately allows for more water storage for plants, thereby resulting in a better T or ET estimate)?"*

**8. Reply:**

We agree with the reviewer that the different approaches have the same objective of increasing the amount of plant-accessible below-ground water storage in order to better reproduce observed transpiration patterns. The refinement of rooting depths (Ichii et al, 2009) and the increase of soil reservoir (de Rosnay and Polcher, 1998) are strategies developed to improve reproduction of observed transpiration rates by increasing the water reservoir. However, the novelty of this manuscript relies on the fact that no other work has addressed this problem with CLM 5 and using at the same time sap flow data to compare in detail the impact of different water reservoirs. Also, we use the plant water stress as an additional indicator of realistic conditions, so we could even detect odd patterns in the model response if we did not have sap flow measurements.

*"Other items:*

*Table 1: Is the p50 correct for the Russian site? I am surprised it would be such a low water potential in such a cold climate."*

**9. Reply:**

We agreed with the reviewer that the default $\Psi_{p50}$ value used for this temperate needle-leaf evergreen site is too low with respect to the values reported for the tree species of the site (from -1.5 Mpa to -3.5 Mpa) (Choat et al, 2012). This point is discussed in the manuscript [Lines: 442-448]. We use deliberately the default configuration for the plant hydraulic stress routine without adjusting the $\Psi_{p50}$ values. This decision was taken to focus only on the impact of modifying the plant water uptake and plant water stress. Despite the low $\Psi_{p50}$ values used for RU-Fyo, the large water availability under the three configurations allows the modeled vegetation to transpire at the potential rate [Lines: 467-468].

Aiming to clarify the decision of using the default configuration of root distribution and plant hydraulic traits, we proposed to add the following in Line 145:

"… et al. 2019). We use the default plant physiological parameterization for each experimental site according to their PFT classification. Site-specific …"

- *"I understand the goal of Figure 4: compare modeled to actual sapflow patterns by time (note that nowhere in the figure or caption is this stated, however). This figure is extremely difficult to comprehend, even after quite a few minutes of study. It is also worth noting that a continuous variable is reported as an area (circle area) rather than a length, leading to potential interpretation ambiguities. Can these not be plotted as regular time series points, whose values vary along a continuous rather than categorical y-axis?"*

**10. Reply:**

Thanks for the recommendation, the simpler the better. Following your recommendation, Figure R2 will replace Figure 4 of the manuscript. This figure change will require to modify the manuscript as follows:

Deleting [Lines: 268-277]:

"… experimental sites. In the graphical scheme adopted in this figure … proportionally to the $\alpha$ value (e.g., ES-Alt.DMC.March)."

Replacing Figure 4 with Figure R2 and updating the figure's caption as follows:

[Figure]

Figure R2. Multi-annual monthly variation of the Pearson correlation coefficient (r) and the index of agreement (Γ) for the default model configuration (DMC), deeper bedrock configuration (DBC), and fractured bedrock configuration (FBC).

*"Figures 5 and A2-A5 are not legible when printed on standard paper and need to be reformatted so that they can be read."*

**11. Reply:**

We will modify these figures to improve their readability.

*"Line 165: It is reported that in order to mimic the hydraulic behaviour of fractured bedrock, it is modeled as a pile of sand (90% sand, 10% clay). This model choice is not supported by any reference to literature on bedrock hydraulic properties, and surprised me as it is not how I would conceive of bedrock hydraulic properties."*

**12. Reply:**

We agree with the reviewer that this is not the way how fractured bedrock properties should be included in the model. We did not address the physical representation of the fractured bedrock, but we used the CLM 5 flexibility to provide the first attempt to

understand the impact of increasing the water reservoir for plants. This aspect is mentioned shortly in methodology [Lines: 166-167], extended with Reply 3 with an addition in [Line: 166], and stressed as well in the conclusions [Lines: 477-480].

**References**

Allard, V., Ourcival, J. M., Rambal, S., Joffre, R., & Rocheteau, A. (2008). Seasonal and annual variation of carbon exchange in an evergreen Mediterranean forest in southern France. Global Change Biology, 14(4), 714-725.

Aranda, I., Pardos, M., Puértolas, J., Jiménez, M. D., & Pardos, J. A. (2007). Water-use efficiency in cork oak (Quercus suber) is modified by the interaction of water and light availabilities. Tree physiology, 27(5), 671-677. https://doi.org/10.1093/treephys/27.5.671

Arneth, A., Kurbatova, J., Kolle, O., Shibistova, O. B., Lloyd, J., Vygodskaya, N. N., & Schulze, E. D. (2002). Comparative ecosystem–atmosphere exchange of energy and mass in a European Russian and a central Siberian bog II. Interseasonal and interannual variability of CO2 fluxes. Tellus B: Chemical and Physical Meteorology, 54(5), 514-530.

Baldocchi, D.D., Ma, S., Rambal, S., Misson, L., Ourcival, J., Limousin, J., Pereira, J. and Papale, D. (2010), On the differential advantages of evergreenness and deciduousness in mediterranean oak woodlands: a flux perspective. Ecological Applications, 20: 1583-1597. https://doi.org/10.1890/08-2047.1

Betsch, P., Bonal, D., Bréda, N., Montpied, P., Peiffer, M., Tuzet, A., & Granier, A. (2011). Drought effects on water relations in beech: the contribution of exchangeable water reservoirs. Agricultural and Forest Meteorology, 151(5), 531-543.

Carcavilla, L., Ruiz, R., and Rodríguez, E. (2008) Guía Geológica del Parque Natural del Alto Tajo. Madrid: Instituto Geológico y Minero de España. 267 p.

Choat, B., Jansen, S., Brodribb, T., Cochard, H., Delzon, S., Bhaskar, R., Bucci, S., Feild, T., Gleason, S., Hacke, U., Jacobsen, A., Lens, F., Maherali, H., Martinez-Vilalta, J., Mayr, S., Mencuccini, M., Mitchell, P., Nardini, A., Pittermann, J., Pratt, R., Sperry, J., Westoby, M., Wright, I., and Zanne, A. (2012) Global convergence in the vulnerability of forests to drought, https://doi.org/10.1038/nature11688

David, T. S., Pinto, C. A., Nadezhdina, N., Kurz-Besson, C., Henriques, M. O., Quilhó, T., ... & David, J. S. (2013). Root functioning, tree water use and hydraulic redistribution in Quercus suber trees: A modeling approach based on root sap flow. Forest Ecology and Management, 307, 136-146. https://doi.org/10.1016/j.foreco.2013.07.012

de Rosnay, P., & Polcher, J. (1998). Modelling root water uptake in a complex land surface scheme coupled to a GCM. Hydrology and Earth System Sciences, 2(2/3), 239-255.

Fan, Y., Miguez-Macho, G., Jobbágy, E. G., Jackson, R. B., & Otero-Casal, C. (2017). Hydrologic regulation of plant rooting depth. Proceedings of the National Academy of Sciences, 114(40), 10572-10577.

Forner, A., Valladares, F., & Aranda, I. (2018). Mediterranean trees coping with severe drought: Avoidance might not be safe. Environmental and experimental botany, 155, 529-540. https://doi.org/10.1016/j.envexpbot.2018.08.006

Gale, M. R., & Grigal, D. F. (1987). Vertical root distributions of northern tree species in relation to successional status. Canadian Journal of Forest Research, 17(8), 829-834. https://doi.org/10.1139/x87-131

Gil-Pelegrín, E., Peguero-Pina, J. J., & Sancho-Knapik, D. (Eds.). (2017). Oaks physiological ecology. Exploring the functional diversity of genus Quercus L. https://doi.org/10.1007/978-3-319-69099-5

Granda, E., Escudero, A., de la Cruz, M., and Valladares, F. (2012) Juvenile–adult tree associations in a continental Mediterranean ecosystem: no evidence for sustained and general facilitation at increased aridity, Journal of Vegetation Science, 23, 164–175, https://doi.org/10.1111/j.1654-1103.2011.01343.x

Granier, A., Ceschia, E., Damesin, C., Dufrêne, E., Epron, D., Gross, P., ... & Saugier, B. (2000a). The carbon balance of a young Beech forest. Functional ecology, 14(3), 312-325.

Granier, A., Biron, P., and Lemoine, D. 2000b. Water balance, transpiration and canopy conductance in two beech stands, Agricultural and Forest Meteorology, 100, 291–308, https://doi.org/10.1016/S0168-1923(99)00151-3

Granier, A., Reichstein, M., Bréda, N., Janssens, I. A., Falge, E., Ciais, P., ... & Wang, Q. (2007). Evidence for soil water control on carbon and water dynamics in European forests during the extremely dry year: 2003. Agricultural and forest meteorology, 143(1-2), 123-145.

Granier, A., Bréda, N., Longdoz, B., Gross, P., & Ngao, J. (2008). Ten years of fluxes and stand growth in a young beech forest at Hesse, North-eastern France. Annals of Forest Science, 65(7), 1.

Grossiord, C., Forner, A., Gessler, A., Granier, A., Pollastrini, M., Valladares, F., & Bonal, D. (2015). Influence of species interactions on transpiration of Mediterranean tree species during a summer drought. European Journal of Forest Research, 134(2), 365-376.

Grillakis, M. G. (2019). Increase in severe and extreme soil moisture droughts for Europe under climate change. Science of The Total Environment, 660, 1245-1255. https://doi.org/10.1016/j.scitotenv.2019.01.001

He, X., Pan, M., Wei, Z., Wood, E. F., and Sheffield, J. 2020. A Global Drought and Flood Catalogue from 1950 to 2016, Bulletin of the AmericanMeteorological Society, 101, E508 – E535, https://doi.org/10.1175/BAMS-D-18-0269.1

Hoff, C., Rambal, S., & Joffre, R. (2002). Simulating carbon and water flows and growth in a Mediterranean evergreen Quercus ilex coppice using the FOREST-BGC model. Forest Ecology and Management, 164(1-3), 121-136. https://doi.org/10.1016/S0378-1127(01)00605-3

Hosseinzadehtalaei, P., Tabari, H., & Willems, P. (2020). Climate change impact on short-duration extreme precipitation and intensity–duration–frequency curves over Europe. Journal of Hydrology, 590, 125249. https://doi.org/10.1016/j.jhydrol.2020.125249

Houston Durrant, T., de Rigo, D., and Caudullo, G. (2016). Fagus sylvatica and other beeches in Europe: distribution, habitat, usage and threats., in: European Atlas of Forest Tree Species, edited by San-Miguel-Ayanz, J., de Rigo, D., Caudullo, G., Houston Durrant, T., and Mauri, A., pp. e014bcd+, Publication Office of the European Union, Luxembourg., https://forest.jrc.ec.europa.eu/en/european-atlas/atlas-download-page/

Hubbert, K. R., Beyers, J. L., & Graham, R. C. (2001). Roles of weathered bedrock and soil in seasonal water relations of Pinus jeffreyi and Arctostaphylos patula. Canadian Journal of Forest Research, 31(11), 1947-1957.

Ichii, K., Wang, W., Hashimoto, H., Yang, F., Votava, P., Michaelis, A. R., & Nemani, R. R. (2009). Refinement of rooting depths using satellite-based evapotranspiration seasonality for ecosystem modeling in California. Agricultural and Forest Meteorology, 149(11), 1907-1918.

Jiang, Z., Liu, H., Wang, H., Peng, J., Meersmans, J., Green, S. M., ... & Song, Z. (2020). Bedrock geochemistry influences vegetation growth by regulating the regolith water holding capacity. Nature communications, 11(1), 1-9.

Jones, D. P., & Graham, R. C. (1993). Water-holding characteristics of weathered granitic rock in chaparral and forest ecosystems. Soil Science Society of America Journal, 57(1), 256-261. https://doi.org/10.2136/sssaj1993.03615995005700010044x

Kirchen, G., Calvaruso, C., Granier, A., Redon, P. O., Van der Heijden, G., Bréda, N., & Turpault, M. P. (2017). Local soil type variability controls the water budget and stand productivity in a beech forest. Forest Ecology and Management, 390, 89-103. https://doi.org/10.1016/j.foreco.2016.12.024

Kurbatova, J., Arneth, A., Vygodskaya, N. N., Kolle, O., Varlargin, A. V., Milyukova, I. M., ... & Lloyd, J. (2002). Comparative ecosystem–atmosphere exchange of energy and mass in a European Russian and a central Siberian bog I. Interseasonal and interannual variability of energy and latent heat fluxes during the snowfree period. Tellus B: Chemical and Physical Meteorology, 54(5), 497-513. https://doi.org/10.3402/tellusb.v54i5.16683

Launiainen, S., Katul, G. G., Kolari, P., Lindroth, A., Lohila, A., Aurela, M., ... & Vesala, T. (2016). Do the energy fluxes and surface conductance of boreal coniferous forests in Europe scale with leaf area?. Global Change Biology, 22(12), 4096-4113. https://doi.org/10.1111/gcb.13497

Lawrence, D. M., Fisher, R. A., Koven, C. D., Oleson, K. W., Swenson, S. C., Bonan, G., Collier, N., Ghimire, B., van Kampenhout, L., Kennedy, D., Kluzek, E., Lawrence, P. J., Li, F., Li, H., Lombardozzi, D., Riley, W. J., Sacks, W. J., Shi, M., Vertenstein, M., Wieder, W. R., Xu, C., Ali, A. A., Badger, A. M., Bisht, G., van den Broeke, M., Brunke, M. A., Burns, S. P., Buzan, J., Clark, M., Craig, A., Dahlin, K., Drewniak, B., Fisher, J. B., Flanner, M., Fox, A. M., Gentine, P., Hoffman, F., Keppel-Aleks, G., Knox, R., Kumar, S., Lenaerts, J., Leung, L. R., Lipscomb, W. H., Lu, Y., Pandey, A., Pelletier, J. D., Perket, J., Randerson, J. T., Ricciuto, D. M., Sanderson, B. M., Slater, A., Subin, Z. M., Tang, J., Thomas, R. Q., Val Martin, M., and Zeng, X.: (2019) The Community Land Model Version 5: Description of New Features, Benchmarking, and Impact of Forcing Uncertainty, Journal of Advances in Modeling Earth Systems, 11, 4245–4287, https://doi.org/10.1029/2018MS001583

Lawrence, D. M., Dagon, K., Kennedy, D., Fisher, R., Sanderson, B. M., Oleson, K. W., ... & Swenson, S. C. (2020). The Community Land Model (CLM5) Parameter Perturbation Ensemble Project: Towards Comprehensive Understanding of Parametric Uncertainty on the Global Terrestrial Carbon Cycle. In AGU Fall Meeting Abstracts (Vol. 2020, pp. B019-0011).

Le Goff, N., & Ottorini, J. M. (2001). Root biomass and biomass increment in a beech (Fagus sylvatica L.) stand in North-East France. Annals of Forest Science, 58(1), 1-13. https://doi.org/10.1051/forest:2001104

Leuschner, C. (2020). Drought response of European beech (Fagus sylvatica L.)—A review. Perspectives in Plant Ecology, Evolution and Systematics, 47, 125576. https://doi.org/10.1016/j.ppees.2020.125576

Limousin, J. M., Rambal, S., Ourcival, J. M., Rocheteau, A., Joffre, R., & Rodriguez-Cortina, R. (2009). Long-term transpiration change with rainfall decline in a Mediterranean Quercus ilex forest. Global Change Biology, 15(9), 2163-2175.

Lüttschwager, D., & Jochheim, H. (2020). Drought primarily reduces canopy transpiration of exposed beech trees and decreases the share of water uptake from deeper soil layers. Forests, 11(5), 537. https://doi.org/10.3390/f11050537

Martín-Duque, R.F., Nicolau, J.M., Martín-Moreno, C., Sánchez, L., Ruiz Lopez de la Cova, R., Sanz, M.A., Lucia, A. (2008). Geomorfologia y gestion del Parque Natural Alto Tajo (1). Condicionantes y criterios geomorfologicos para la restauracion de minas de caolin. Trabajos de Geomorfologia en Espana. X Reunion Nacional de geomorfologia. Cadiz. http://www.landformining.igeo.ucm-csic.es/sites/default/files/files/Mart%C3%ADn%20Duque%20et%20al,%202008_AT(1).pdf

Martín-Moreno, C., Fidalgo Hijano, C., Martín Duque, J., González Martín, J., Zapico Alonso, I., and Laronne, J. (2014) The Ribagorda sand gully (east-central Spain): Sediment yield and

human-induced origin, Geomorphology, 224, 122–138, https://doi.org/10.1016/j.geomorph.2014.07.013

Maxwell, R. M., & Condon, L. E. (2016). Connections between groundwater flow and transpiration partitioning. Science, 353(6297), 377-380.

McCormick, E. L., Dralle, D. N., Hahm, W. J., Tune, A. K., Schmidt, L. M., Chadwick, K. D., & Rempe, D. M. (2021). Widespread woody plant use of water stored in bedrock. Nature, 597(7875), 225-229.

Milyukova, I. M., Kolle, O., Varlagin, A. V., Vygodskaya, N. N., Schulze, E. D., & Lloyd, J. (2002). Carbon balance of a southern taiga spruce stand in European Russia. Tellus B: Chemical and Physical Meteorology, 54(5), 429-442.

Mu, M., De Kauwe, M. G., Ukkola, A. M., Pitman, A. J., Gimeno, T. E., Medlyn, B. E., Or, D., Yang, J., and Ellsworth, D. S. (2021) Evaluating a land surface model at a water-limited site: implications for land surface contributions to droughts and heatwaves, Hydrol. Earth Syst. Sci., 25, 447–471, https://doi.org/10.5194/hess-25-447-2021

Nadezhdina, N., Ferreira, M. I., Silva, R., & Pacheco, C. A. (2008). Seasonal variation of water uptake of a Quercus suber tree in Central Portugal. Plant and Soil, 305(1), 105-119. https://doi.org/10.1007/s11104-007-9398-y

Novenko, E. Y., & Zuganova, I. S. (2010). Landscape dynamics in the Eemian interglacial and Early Weichselian glacial epoch on the South Valdai Hills (Russia). The Open Geography Journal, 3(1).

Ollivier, C., Olioso, A., Carrière, S. D., Boulet, G., Chalikakis, K., Chanzy, A., ... & Weiss, M. (2021). An evapotranspiration model driven by remote sensing data for assessing groundwater resource in karst watershed. Science of the Total Environment, 781, 146706.

Pelletier, J. D., Broxton, P. D., Hazenberg, P., Zeng, X., Troch, P. A., Niu, G.-Y., Williams, Z., Brunke, M. A., and Gochis, D. (2016) A gridded global data set of soil, intact regolith, and sedimentary deposit thicknesses for regional and global land surface modeling, Journal of Advances in Modeling Earth Systems, 8, 41–65, https://doi.org/10.1002/2015MS000526

Penuelas, J., & Filella, I. (2003). Deuterium labelling of roots provides evidence of deep water access and hydraulic lift by Pinus nigra in a Mediterranean forest of NE Spain. Environmental and Experimental Botany, 49(3), 201-208.

Pita, G., Gielen, B., Zona, D., Rodrigues, A., Rambal, S., Janssens, I. A., & Ceulemans, R. (2013). Carbon and water vapor fluxes over four forests in two contrasting climatic zones. Agricultural and forest meteorology, 180, 211-224.

Rempe, D. M. and Dietrich, W. E.: (2018) Direct observations of rock moisture, a hidden component of the hydrologic cycle, Proceedings of the National Academy of Sciences, 115, 2664–2669, https://doi.org/10.1073/pnas.1800141115

Špulák, O., Šach, F., & Kacálek, D. (2021). Topsoil Moisture Depletion and Recharge below Young Norway Spruce, White Birch, and Treeless Gaps at a Mountain-Summit Site. Forests, 12(7), 828. https://doi.org/10.3390/f12070828

Schulze, E. D., Vygodskaya, N. N., Tchebakova, N. M., Czimczik, C. I., Kozlov, D. N., Lloyd, J., ... & Wirth, C. (2002). The Eurosiberian Transect: an introduction to the experimental region. Tellus B: Chemical and Physical Meteorology, 54(5), 421-428.

Vygodskaya, N. N., Schulze, E. D., Tchebakova, N. M., Karpachevskii, L. O., Kozlov, D., Sidorov, K. N., ... & Pugachevskii, A. V. (2002). Climatic control of stand thinning in unmanaged spruce forests of the southern taiga in European Russia. Tellus B: Chemical and Physical Meteorology, 54(5), 443-461.

Zapater, M., Hossann, C., Bréda, N., Bréchet, C., Bonal, D., & Granier, A. (2012). Evidence of hydraulic lift in a young beech and oak mixed forest using 18O soil water labelling. Trees, 25(5), 885-894. https://doi.org/10.1007/s00468-011-0563-9

Zeng, Y., Xie, Z., Liu, S., Xie, J., Jia, B., Qin, P., & Gao, J. (2018). Global land surface modeling including lateral groundwater flow. Journal of Advances in Modeling Earth Systems, 10(8), 1882-1900.

---

## Author Comment (AC2)

**Reply to comments of Reviewer 1**

In **_"blue"_** we copied the comments of the reviewer, in **black** our reply, in **green** new additions that will be included in the manuscript, and in **red** the deletions to the manuscript. We also indicated within square brackets the line number when we refer to specific points or sections of the manuscript "[Line ]". The numbering of figures and tables of this reply are preceded by R aiming to differentiate them from the original figures and tables of the manuscript.

_"**Summary:**_
_The authors manipulate existing CLM 5 soil texture and depth to bedrock (DTB) parameters to account for plant-accessible moisture stored in fractured bedrock beneath soils. They compare simulated to actual transpiration across four sites in Europe with different climate conditions and find that the two simulated bedrock scenarios (one with 1.5 m of simulated "bedrock" and another with additional "fracturing") better match observed transpiration during the summer for the sites with a pronounced dry season._

_This work motivates further exploration of how bedrock water is accessed by plants and how this process is represented in hydrologic and ecophysiological models. The manuscript is exceptionally well motivated and contextualized, and was easy to follow. The conclusions are well reasoned and of interest to a number of communities engaged in biogeosciences research. Comments are shared to increase clarity."_

**1. Reply:**

We would like to thank the reviewer's for the positive comments and constructive feedback. Hereafter, we provide a separate response to each comment to clarify the points raised by the reviewer and improving the manuscript.

_**Comments and questions for the authors:**_

_Why are the rooting profiles illustrated as linear but described as exponential?_

**2. Reply:**

The rooting profiles showed in Figure 1 provide a visual representation of the roots distribution and were not intended to provide a mathematical representation of the exponential rooting profile. However, to prevent any misinterpretation for the readers, we improved the graphics of this figure and changed it from linear to an exponential representation (Figure R1). Consequently, Figure R1 will replace Figure 1 in the manuscript keeping the caption unchanged.

[Figure]

Figure R1. Modified version of the vertical distribution profile of roots according to the exponential function used by CLM 5. This figure will replace Figure 1 of the manuscript.

*Is there an expectation that model agreement should be improved during energy limited periods or in energy limited sites (e.g. line 303, 405). Limitations outside of water availability could be better quantified and described in the results and discussion.*

**3. Reply:**

Yes, the expectation is that any of the model configurations should agree with the observed transpiration under energy-limited conditions. Nonetheless, under these conditions, we expect that the modeled vegetation does not deplete the soil water reservoir, which is the case in these sites. Due to the lack of plant water stress, the model configurations can transpire at maximum capacity during winter (for the evergreen sites), spring, and autumn [Lines 323-324]. This response during unstressed periods happens because the precipitation water is enough to replenish the transpired water [Lines 376-

377, 420-421] as it happens in FR-Hes during summertime. In this regard, water availability (surplus and deficit) is the main driving factor for the differences among model configurations that only include the changes to subsurface conditions.

The manuscript mentioned that atmospheric forcing [Lines 223-226] and plant physiological parameters [Lines 144-145] do not change among models. Also, we discussed the role of plant hydraulics on the water movement within the plant [Lines 417-420, 442-447]. However, we cannot discard the possibility that the increased water stress in summertime could result from an over-estimation of water use during the early season that depletes the water reservoirs. Under these conditions, the overestimation could be linked to a lack of water use strategy or additional stressors not accounted for in the model formulation.

Consequently, we propose to add:

In [Line 428]:

"… stems and leaves (Fig 5.). Also, the increased water stress in summertime could result from an over-estimation of water use during the early season that depletes the water reservoirs. Under these conditions, the overestimation indicates a lack of water use strategy or additional stressors not accounted for in the default model parameterization."

*Are the default parameters reported in line 145 site specific and if so, are they reported?*

**4. Reply:**

No. None of the default parameters for root distribution and plant hydraulics were adjusted to site conditions. Also, the parameters are available in Table 1.

*A statement on the rationale behind the specific three model configurations would benefit the reader. Why these three and not other possibilities? Additionally, in line 167, how does the 90% sand and 10% clay mimic fractured bedrock? Justification is needed here.*

**5. Reply:**

We agreed with the reviewer's comment that additional information will benefit the readers. The selection of these model configurations was done aiming to provide a straight evaluation of the impact of changing the depth to bedrock and soil properties. In this manuscript, we used the measured transpiration as benchmark to compare the model transpiration, however, the Default Model Configuration (DMC) was used as a baseline to compare the effects in the simulated percent loss of conductivity (PLC).

The Deeper Bedrock Configuration (DBC) shifts the DTB adding a predetermined profile of 1.5 m below the DTB in the DMC. This depth was defined as a fixed parameter across all sites because the heterogeneous nature of weathered bedrock depends on the interaction between climate, vegetation, and rock type (Pawlik *et al.* 2016).

Moving the bedrock deeper implies solving the issue of how to parameterize this newly created water storage. "Section 3.2 Bedrock Configuration" [Line: 154] describes the selected approach which relies on two main assumptions: a fractured bedrock will have a high-water conductivity and low water holding capacity. This approach is based on the recent studies at regional and continental scales stressing the importance of the accumulated sediments in weathered bedrock fractures that enhances the water holding capacity beneath the soil (Jian *et al.* 2020), allowing the woody ecosystems to use the rock moisture at regional scales (McCormick *et al.* 2021). The soil texture selection that mimics a fractured bedrock is supported by the similarities in saturated hydraulic conductivity ($K_s$) that both, sandy soils (Miyazaki, 1996; Pachepsky and Park, 2015) and weathered granite rocks have (Rouxel et al., 2010; Katsura et al., 2009) [Lines: 396-400]. As the sandy soil texture classification can be described by any soil with a combination of more than 85% of sand and less than 10% of clay, we decided to choose the combination of soil fractions that provides a sandy soil texture with a maximum water holding capacity for this textural class.

Consequently, we propose to add:

In [Line 163]:

"... each location, using this model configuration as a baseline for comparison across the manuscript. The second ..."

In [Line 164]:

"... texture classification. This depth was defined as a fixed parameter across all sites because the thickness and degree of bedrock weathering are difficult to characterize over broad scales (Holbrook et al. 2014), where the interaction between climate, vegetation, and rock type determines the extent and properties of the weathered bedrock (Pawlik et al. 2016). The third configuration ..."

In [Line 166]:

"... clay (Table 1). This approach aims to mimic the hydrological behavior of a fractured bedrock based on two main assumptions: a fractured bedrock should have a high-water conductivity and low water holding capacity. As the sandy soil texture classification can be described by any soil with a combination of more than 85% of sand and less than 10% of clay, we decided to choose the combination of soil fractions that provides a sandy soil texture with a maximum water holding capacity for this textural class. The high sand percentage will mimic the fast water movement through the primary and secondary porosity of the fractured bedrock. At the same time, the low clay content allows having a low water holding capacity for plant water uptake compared with the above soil layers."

*The Pelletier et al 2016 dataset is a model output and not reflective of local site conditions per se. As far as I understand, it is only validated for depth to bedrock in the US using groundwater well data (which is rarely available in uplands areas like the sites in this study.) The language around use of the dataset should be couched to reflect that the dataset does not provide DTB at the four sites.*

**6. Reply:**

Yes, the data set from Pelletier et al (2016) was validated for US and then applied to estimate the depth to bedrock (DTB) globally. However, the differences between the DTB used in the model and that reported for the experimental sites does not show differences larger than 50 cm for FR-Pue and FR-Hes, and in ES-Alt it is within the reported field data (Table R1). In RU-Fyo the first 50 cm are reported as peat, lying above glacial deposits (Table R2).

Table R1. Differences between the soil depths ingested in the model and the soil depths reported for the sites.

|  | Experimental Sites | | | |
| --- | --- | --- | --- | --- |
|  | ES-Alt | FR-Pue | FR-Hes | RU-Fyo |
| Input for the Model [cm] | 82 cm | 95 cm | 101 cm | 132 cm |
| Reported Field Data [cm] | 20-100cm | 50 cm | 145 cm | 50 cm (peat) |
| References | Grossiord et al. (2015) Martin-Duque et al. (2008) Penuelas et al. (2003) | Pita et al. (2013) | Granier et al. (2000b) | Arneth et al. (2002) Kurbatova et al. (2002) Schulze et al. (2002) Vigodskaya et al. (2002) |

Following the reviewer's recommendation, we will add the following:

In [Line 149]:

"… summary information. Although this data set does not reflect the exact site conditions, the depth to bedrock is closer to measurements reported in previous studies at the experimental sites. The simulations were …"

Also, we will add Table R2 as an appendix describing the reported data for each experimental site.

Table R2. Summary of the main sub-surface characteristics of each experimental site.

| Main Site Characteristics | ES-Alt | FR-Pue | FR-Hes | RU-Fyo |
| --- | --- | --- | --- | --- |
| Rooting Depth (m) | 8.0 | 4.5 | > 1.5 | 0.2 |
| Peat Layer Depth (cm) | None | None | None | 50 |
| Soil Depth (cm) | 20-40 \| 100 | 50 | 145 | N.A. |
| Soil Texture: % Clay | 25.9 | 39 | 25 | N.A. |
| Soil Texture: % Sand | 57.3 | 26 | N.A. | N.A. |
| Soil Texture: % Silt | 16.8 | 35 | N.A. | N.A. |

| Soil Type | Sandy Clay Loam | Silty Clay Loam | Clay Loam | Loam |
|---|---|---|---|---|
| Soil Permeability | N.A. | High | N.A. | N.A. |
| Superficial Stone Fraction | N.A. | 0.75 (0-50cm) | N.A. | N.A. |
| Deep Stone Fraction | N.A. | 0.90 (>50 cm) | N.A. | N.A. |
| Bedrock Type | Cretaceous carbonate | Jurassic Limestone | Sandstone | Glacial deposits |
| Water Table Depth | Deep | Deep | Deep | Superficial |
| Plant Water | Soil Water, Weathered Bedrock, Groundwater | Soil Water, Weathered Bedrock, Groundwater | Soil Water | Non-Saturated Substratum |
| Hydraulic Lift | Present | N.A. | Present | N.A. |
| References | Carcavilla et al. (2008) Forner et al. (2018) Grossiord et al. (2015) Martin-Duque et al. (2008) Penuelas et al. (2003) | Allard et al. (2008) Limousin et al. (2009) Pita et al. (2013) | Betsch et al. (2011) Granier et al. (2007) Granier et al. (2000a) Granier et al. (2000b) Le Goff et al. (2001) Zapater et al. (2012) | Arneth et al. (2002) Kurbatova et al. (2002) Milyukova et al. (2002) Novenko et al. (2010) Schulze et al. (2002) Vigodskaya et al. (2002) |

*In line 80, what does "fully developed" mean in this context?*

**7. Reply:**

The term "fully developed" was intended to convey the description of a canopy showing the maximum leaf area index (LAI) during the growing season. However, we see that this may misguide the reader. Consequently, we propose to change it as follows [Line 80]:

"… It is expected that during summer, the vegetation has a fully developed canopy exhibiting a peak in leaf area index, extracting soil water at the maximum rate, and relying on deeper soil water pools during extended dry periods."

*In line 75, an additional possibility is that belowground biomass distributions may change over time in response to water stress (e.g. Liu et al, 2019).*

**8. Reply:**

Good point, thank you. Following this recommendation, we will add the following in [Line 74]:

"… This latter approach emphasizes the key role played by soil texture and rock fragments in regulating the response of root growth (Hu *et al.,* 2021, Li et al., 2019) and plant hydraulics to soil drying …"

*Is there site specific subsurface information (from e.g. the papers cited in the site descriptions) that could be added to contextualize the DTB increase needed to improve model performance?*

**9. Reply:**

Yes, there is available information in Table R2 that will be added as an appendix (see reply 6). Also, we propose to add the following lines to improve the site descriptions:

In [Line: 95]:

"… and lowlands (RU-Fyo). All the sites lack detailed information concerning the physical properties of the weathered bedrock, such as hydraulic conductivity, water holding capacity, or percentage of fractures. The sites are distributed across an environmental …"

For ES-Alt [Line: 101]:

"… et al.  2017). The soils are formed from Cretaceous carbonate rocks settled on top of sandy sediments (Carcavilla et al.  2008), showing a poor development (Granda et al. 2012) with maximum soil thickness varying between 25 cm at the top of the slopes, and more than 1.0 m at the bottom (Martín-Moreno et al.  2014). The climate …"

[Line: 108]:

"… et al.  2014). Quercus ilex trees at this site have the capacity to allocate structural and fine roots down to 8.0 m depth (Penuelas et al.  2003). The plant …"

For FR-Pue [Line: 112]:

"… et al.  2002). The superficial soil layers have a high soil permeability thanks to the elevated stone fraction of 0.75. On the other hand, soil layers beneath 50 cm have a larger clay content (> 30%) and a stone fraction of more than 0.9 (Limousin et al.  2009, Pita et al.  2013). This location …"

[Line: 115]:

"… 2008). This forest stand allocates most of the fine root biomass in the top 50 cm of the soil profile with a small fraction of roots reaching depths down to 4.5 m (Allard et al. 2008). Similarly, to …"

For FR-Hes [Line: 112]:

"… 2007). It is a clear transition between the eluviated horizon and the horizon with high clay accumulation at 50 cm depth (Zapater et al.  2012), leading to a low clay content in the upper soil layers (Granier et al.  2000 a). It has … "

[Line: 124]:

"… 2001). Fagus sylvatica trees allocate most of the fine roots in the upper 40 cm of soil with some fine roots reaching depths down to 1.5 m (Granier et al. 2000a, Granier et al. 2000b, Zapater et al.  2012, Bestch et al. 2011). The dominant …"

For RU-Fyo [Line: 131]:

"… and still present. The peat layer at this site has an average depth of 50 cm (Schulze et al. 2002, Vygoskaya et al. 2002, Arneth et al. 2002, Kurbatova et al. 2002), and the glacial deposits result in a loamy texture of the soil beneath it (Novenko et al. 2010, Schulze et al. 2002). The water table at this site is shallow, forcing the trees to allocate most of the fine roots in the top 20 cm of the soil (Milyukova et al. Schulze et al. 2002, Vygodskaya et al. 2002). The climate …"

*The overprediction of transpiration during spring and rapid drying of the root zone is a very interesting result that models representing deeper water stores will have to grapple with. The discussion of plant hydraulics in L420-440 is thorough and very well done, but are there perhaps other additional factors that could be considered as well? For example, dynamic belowground biomass, fungi, or the role of multi-porosity systems (e.g. Schwinning, 2020).*

**10. Reply:**

The additional factors mentioned by the reviewer may help to explain the measured transpiration during the dry season at sites such as ES-Alt and FR-Pue, where the precipitation is reduced, and plants must access any source of water available belowground to survive during this period. In this context, the four potential pathways proposed by Schwinning (2020) could play an important role for this vegetation.

We propose to add the following in [Line 436]:

"… vegetation types. Also, plants depending on rock moisture had developed special water access strategies such as dynamic root systems or mycorrhizae growing along the rock cracks and accessing water stored in or dripping from the bedrock (Schwinning 2020). Experimental evidence …"

*Is it necessary to have well developed soil to access groundwater (Line 355)?*

**11. Reply:**

No, it is not necessary to have well developed soil to access groundwater. Nonetheless, to improve the idea we want to convey, we will add the following lines:

In [Line 354]:

"… If geological conditions allow the formation of deep soils (e.g., Amazon Basin, Loess Plateau), the roots will access deep groundwater that will become very

important for surviving extended dry periods (Chitra-Tarak et al., 2021; Tao et al., 2021). On the other hand, when soils are shallow and less developed, the trees must thrive by accessing additional water pockets in the weathered bedrock. The heterogeneous nature of weathered bedrock depends on the interaction between climate, vegetation, and rock type (Pawlik et al. 2016). These interactions allow the increment of the water-holding capacity of weathered bedrock by increasing the porosity and mineral surface area (Navarre-Sitchler et al. 2015). This water holding capacity is considered negligible (Novák & Šurda, 2010), but the vertical extent of this layer makes the water reservoir large enough to support deep rooting vegetation during dry spells (Graham et al. 2010, Jones & Graham 1993). As an example, Mediterranean trees are able to uptake water from the deep vadose zone …"

*The definition of bedrock within the paper is a bit inconsistent, specifically in the caption of Figure 1. For example, bedrock in CLM5 is considered impermeable but bedrock is represented as a combination of sand and clay. Clarification is needed here.*

**12. Reply:**

It is true that in CLM 5 the bedrock is described as an impermeable layer and the paper also states that in [Line 158]. Also, in Figures 1B, 1C, and 1D the impermeable bedrock layer is represented by a solid block. The term "Fractured Bedrock Configuration" that is in the caption of Figure 1 is explained in the methodology [Lines 164-167] and the description is extended with the proposed addition in reply 5. To improve clarity further, we will change the caption text to:

"Figure 1. Geographical location of the experimental sites and spatial distribution of the depth to bedrock across Europe (A) based on Pelletier et al. (2016). The graphics below the map are the schematic of the three model configurations used in this work: default model configuration, DMC (B); deeper bedrock configuration, DBC (C); and fractured bedrock configuration, FBC (D). The block with dashed lines represents the soil profile, the solid grey block represents the impermeable bedrock layer as it is assumed by CLM 5, and the translucid grey block represents the mimicked fractured bedrock layer."

*Some comments about figures:*

*Figure A1: Is this a boxplot? It seems like a timeseries. A description of the points vs. lines is needed in the caption.*

**13. Reply:**

Yes, those are boxplots showing the multiannual variability per day of the year. The points are the outliers for each boxplot when present. We will improve the figure caption as follows:

> "Figure A1. Multi-annual daily boxplots for potential evaporation ($E_o$), stand transpiration ($E_T$), and vapor pressure deficit ($\Lambda$) of the selected experimental sites across Europe. The boxplot represents the data contained between the first and third quartiles, the central line is the median, the whiskers represent a predefined distance from the median (1.5 x inter-quartile range), and the dots are the outliers."

*Figure 2: Is there a legend label missing (corresponding to pink or orange)?*

**14. Reply:**

There is actually not a missing label but the bars with a pink looking color are translucid red representing $E_o$ that it is solid red in the legend. We will improve the figure as follows:

[Figure]

Figure R2. Updated version of Figure 2 to be replaced in the manuscript.

*Figure 3: This is the most impactful figure but it is very difficult to tell the different model configurations apart. The caption says boxes but there don't seem to be boxes in the figure.*

**15. Reply:**

Thanks for the heads-up on this. This plot shows only the boxplot (inter-quartile range), but it is difficult to distinguish between model configurations. Consequently, we will replace Figure 3 and caption as follows:

[Figure]

Figure R3. Median of the measured and modeled daily transpiration rates for the different model configurations at each experimental site.

*Figure 4: Some of the concepts in this figure could potentially be better represented by scatterplots for specific times or model configurations that are most significant to the results. Including a simple illustration of how model-data agreement is improved with bedrock water storage under specific conditions could make the paper potentially more impactful and approachable to non-CLM experts. "*

**16. Reply:**

Thanks for the recommendation, the simpler the better. We decided to change this figure into an annual time series (Figure R2) that will replace Figure 4 of the manuscript. This figure change will require to modify the manuscript as follows:

Deleting [Lines: 268-277]:

"... experimental sites. In the graphical scheme adopted in this figure ... proportionally to the $\alpha$ value (e.g., ES-Alt.DMC.March)."

Replacing Figure 4 with Figure R2 and updating the figure's caption as follows:

[revised manuscript text omitted]

Limousin, J. M., Rambal, S., Ourcival, J. M., Rocheteau, A., Joffre, R., & Rodriguez-Cortina, R. (2009). Long-term transpiration change with rainfall decline in a Mediterranean Quercus ilex forest. Global Change Biology, 15(9), 2163-2175.

Li, H., Si, B., Wu, P., & McDonnell, J. J. (2019). Water mining from the deep critical zone by apple trees growing on loess. Hydrological Processes, 33(2), 320-327. https://doi.org/10.1002/hyp.13346

Martín-Duque, R.F., Nicolau, J.M., Martín-Moreno, C., Sánchez, L., Ruiz Lopez de la Cova, R., Sanz, M.A., Lucia, A. (2008). Geomorfologia y gestion del Parque Natural Alto Tajo (1). Condicionantes y criterios geomorfologicos para la restauracion de minas de caolin. Trabajos de Geomorfologia en Espana. X Reunion Nacional de geomorfologia. Cadiz. http://www.landformining.igeo.ucm-csic.es/sites/default/files/files/Mart%C3%ADn%20Duque%20et%20al,%202008_AT(1).pdf

Martín-Moreno, C., Fidalgo Hijano, C., Martín Duque, J., González Martín, J., Zapico Alonso, I., and Laronne, J. (2014) The Ribagorda sand gully (east-central Spain): Sediment yield and human-induced origin, Geomorphology, 224, 122–138, https://doi.org/10.1016/j.geomorph.2014.07.013

McCormick, E. L., Dralle, D. N., Hahm, W. J., Tune, A. K., Schmidt, L. M., Chadwick, K. D., & Rempe, D. M. (2021). Widespread woody plant use of water stored in bedrock. Nature, 597(7875), 225-229.

Miyazaki, T.: Bulk density dependence of air entry suctions and saturated hydraulic conductivities of soils, Soil Science, 161, 484–490, 1996.

Milyukova, I. M., Kolle, O., Varlagin, A. V., Vygodskaya, N. N., Schulze, E. D., & Lloyd, J. (2002). Carbon balance of a southern taiga spruce stand in European Russia. Tellus B: Chemical and Physical Meteorology, 54(5), 429-442.

Navarre-Sitchler, A., Brantley, S. L., & Rother, G. (2015). How porosity increases during incipient weathering of crystalline silicate rocks. Reviews in Mineralogy and Geochemistry, 80(1), 331-354.

Novák, V., & Šurda, P. (2010). The water retention of a granite rock fragments in High Tatras stony soils. Journal of Hydrology and Hydromechanics, 58(3), 181-187.

Novenko, E. Y., & Zuganova, I. S. (2010). Landscape dynamics in the Eemian interglacial and Early Weichselian glacial epoch on the South Valdai Hills (Russia). The Open Geography Journal, 3(1).

Pachepsky, Y. and Park, Y.: Saturated Hydraulic Conductivity of US Soils Grouped According to Textural Class and Bulk Density, Soil Science Society of America Journal, 79, 1094–1100, https://doi.org/10.2136/sssaj2015.02.0067, 2015.

Pawlik, Ł., Phillips, J. D., and ˘Samonil, P. (2016) Roots, rock, and regolith: Biomechanical and biochemical weathering by trees and its impact on hillslopes—A critical literature review, Earth-Science Reviews, 159, 142–159, https://doi.org/10.1016/j.earscirev.2016.06.002

Pelletier, J. D., Broxton, P. D., Hazenberg, P., Zeng, X., Troch, P. A., Niu, G.-Y., Williams, Z., Brunke, M. A., and Gochis, D. (2016) A gridded global data set of soil, intact regolith, and sedimentary deposit thicknesses for regional and global land surface modeling, Journal of Advances in Modeling Earth Systems, 8, 41–65, https://doi.org/10.1002/2015MS000526

Penuelas, J., & Filella, I. (2003). Deuterium labelling of roots provides evidence of deep water access and hydraulic lift by Pinus nigra in a Mediterranean forest of NE Spain. Environmental and Experimental Botany, 49(3), 201-208.

Pita, G., Gielen, B., Zona, D., Rodrigues, A., Rambal, S., Janssens, I. A., & Ceulemans, R. (2013). Carbon and water vapor fluxes over four forests in two contrasting climatic zones. Agricultural and forest meteorology, 180, 211-224.

Rouxel, M., Molénat, J., Ruiz, L., Chirié, G., and Hamon, Y.: Determination of saturated and unsaturated hydraulic conductivities and water retention curves of weathered granite, in: EGU General Assembly Conference Abstracts, EGU General Assembly Conference Abstracts, p. 3623, 2010.

Schwinning, S. (2020) A critical question for the critical zone: how do plants use rock water?. Plant Soil 454, 49–56. https://doi.org/10.1007/s11104-020-04648-4

Schulze, E. D., Vygodskaya, N. N., Tchebakova, N. M., Czimczik, C. I., Kozlov, D. N., Lloyd, J., ... & Wirth, C. (2002). The Eurosiberian Transect: an introduction to the experimental region. Tellus B: Chemical and Physical Meteorology, 54(5), 421-428.

Vygodskaya, N. N., Schulze, E. D., Tchebakova, N. M., Karpachevskii, L. O., Kozlov, D., Sidorov, K. N., ... & Pugachevskii, A. V. (2002). Climatic control of stand thinning in unmanaged spruce forests of the southern taiga in European Russia. Tellus B: Chemical and Physical Meteorology, 54(5), 443-461.

Zapater, M., Hossann, C., Bréda, N., Bréchet, C., Bonal, D., & Granier, A. (2012). Evidence of hydraulic lift in a young beech and oak mixed forest using 18O soil water labelling. Trees, 25(5), 885-894. https://doi.org/10.1007/s00468-011-0563-9

---

## Author Response (AR1)

Dear Editor,

This document shows a detailed description of all major changes applied to the manuscript and catalogued in two sections. All changes are preceded by "C" and numbered continuously, also indicating the line number of the original manuscript where the change is being included, where additions are colored with **green** and deletions with **red**. Section 1 contains additional changes that all authors considered of highly importance to be added to the manuscript and the justification of why to include it. Section 2 contains all major changes coming from the reply to reviewers without additional modifications from the authors.

**Section 1**

C1. The authors want to make clear the message that the inclusion of a weathered bedrock in the Community Land Model (CLM 5) is aiming to show the importance of considering the bedrock as an active hydrological component from which vegetation also relay on. We are not claiming improvements on the overestimation of spring transpiration on the drier sites, because it is shown that this issue is linked to the control of a plant water use strategy, where the soil and rock moisture reservoirs are not part of.

Addition in [Line 480]:

"... Fig. 5). However, the extreme water stress in summer time in the DMC configuration could have resulted from an overestimation of water use during the early season, leading to depleted water reservoirs in summer. Although the DBC and FBC configurations improved the simulated transpiration and water stress experienced by the stem and leaves during summer by increasing the soil water availability, neither of the configurations reduced the transpiration in spring. This could suggest that the reason for the underestimated summer transpiration might not be an underestimated water storage capacity but some other model deficiency that is responsible for both, overestimated transpiration in spring and subsequently underestimated transpiration in summer. "

C2. On the reply to Reviewer #1 (Reply 11), we agreed on modifying Figure 5 to improve the readability but we did not include the modified figure because it took some time to decide the better layout. Now, we are including the new figure and the respective caption:

Figure 5. Hydraulic stress experienced by the modeled vegetation per experimental site based on the vulnerability curves of each plant organ (See Fig. A2 to Fig. A5) between June and October. Each plot describes the distribution of the hydraulic stress experienced by stem xylem and leaves xylem expressed as percentage of conductivity ( $\Xi$ ).

**Section 2**

C3. Addition in [Line: 33]:

[revised manuscript text omitted]

C17. Addition in [Line: 309]:

"... configurations.

The soil parametrization used in DMC and DBC agreed with the published data for FR-Hes, FR-Pue, and FR-Hes (Figure A2). These configurations are located within the boundaries of the soil texture classification of each site. In this regard, the agreement of soil water storage

capacity, infiltration, and percolation rates agreed with the expected site conditions. Clay content in ES-Alt is similar between model configurations and site conditions, providing a similar water holding capacity. However, the sand content of the model for DMC and DBC configurations represents a lower hydraulic conductivity than the one expected for this site."

**C18. Addition in [Line 395]:**

[revised manuscript text omitted]

C23. In Figure 1, we modified the root profile shape to match the exponential pattern explained along the manuscript. Also, we improve the caption accordingly.

---

## Author Response (AR2)

**Reply to Editor's minor revision**

In **blue** we copied the comments of the editor, in **black** our reply, in **green** new additions that will be included in the manuscript, and in **red** the deletions to the manuscript. We also indicated within square brackets the line number when we refer to specific points or sections of the manuscript "[Line: ]". The numbering of figures and tables of this reply are preceded by R aiming to differentiate them from the original figures and tables of the manuscript.

Comments to the author:
Dear Authors,
Happy to let you know that I find this paper ready for publication after minor modifications for two figures:
**1. Reply:**
The authors are placed with this new manuscript and your support as Associate Editor, we appreciate your time and effort.

Figure 2 caption is unclear: Does the precipitation (P), transpiration (Et), and the potential evapotranspiration (Eo) values, are monthly sums? And is the air temperature (T) of the monthly average?
**2. Reply:**
Aiming to clarify the caption of Figure 2, we updated it as follows:
"Figure 2. Temporal variation of the maximum vapor pressure deficit (VPD), total precipitation ($P$), total potential evaporation ($E_o$), total transpiration ($E_T$), and mean air temperature ($T$) for the selected experimental sites across Europe. The monthly values are based on the different sampling periods for each site."

Figure 5: Please explain the figures' shapes, i.e., the meanings of their gray and green bodies' widths and lengths (distribution and frequency)? And of the horizontal line (monthly mean?).
**3. Reply:**
Aiming to improve the readability of Figure 5, we update its caption as follows:
"Figure 5. Hydraulic stress experienced by the modeled vegetation per experimental site based on the vulnerability curves of each plant organ (See Fig. A1 to Fig. A6) between June and October. Each vioplot describes the distribution of the hydraulic stress experienced by stem xylem (X) and sunny leaves xylem (L) expressed as percentage of conductivity ($\Xi$). Each vioplot is a visual representation of the probability density of the data, with the width and length representing the frequency and data distribution, respectively. The horizontal line corresponds to the median value and the thicker vertical line represents the range between first and third quartiles of the data."

I wish you the best in your future work and kind regards, Eyal
**4. Reply:**
Thanks for your support.